# Abortive intussusceptive angiogenesis causes multi-cavernous vascular malformations

**Wenqing Li[1], Virginia Tran[1], Iftach Shaked[2], Belinda Xue[1], Thomas Moore[3], Rhonda Lightle[3], David Kleinfeld[2,4], Issam A Awad[3], Mark H Ginsberg[1]\***

[1]Department of Medicine, University of California, San Diego, La Jolla, United States; [2]Department of Physics, University of California, San Diego, La Jolla, United States; [3]Neurovascular Surgery Program, Section of Neurosurgery, Department of Surgery, University of Chicago School of Medicine and Biological Sciences, Chicago, United States; [4]Section of Neurobiology, University of California San Diego, La Jolla, United States

**Abstract** Mosaic inactivation of *CCM2* in humans causes cerebral cavernous malformations (CCMs) containing adjacent dilated blood-filled multi-cavernous lesions. We used CRISPR-Cas9 mutagenesis to induce mosaic inactivation of zebrafish *ccm2* resulting in a novel lethal multi-cavernous lesion in the embryonic caudal venous plexus (CVP) caused by obstruction of blood flow by intraluminal pillars. These pillars mimic those that mediate intussusceptive angiogenesis; however, in contrast to the normal process, the pillars failed to fuse to split the pre-existing vessel in two. Abortive intussusceptive angiogenesis stemmed from mosaic inactivation of *ccm2* leading to patchy *klf2a* overexpression and resultant aberrant flow signaling. Surviving adult fish manifested histologically typical hemorrhagic CCM. Formation of mammalian CCM requires the flow-regulated transcription factor KLF2; fish CCM and the embryonic CVP lesion failed to form in *klf2a* null fish indicating a common pathogenesis with the mammalian lesion. These studies describe a zebrafish CCM model and establish a mechanism that can explain the formation of characteristic multi-cavernous lesions.

**\*For correspondence:**
mhginsberg@ucsd.edu

**Competing interests:** The authors declare that no competing interests exist.

## Introduction

Cerebral cavernous malformations (CCMs) are central nervous system (CNS) vascular anomalies that lead to significant morbidity and mortality (*Leblanc et al., 2009*). CCMs affect ~1/200 humans and cause a lifelong risk of stroke and other neurological sequelae for which there is no pharmacological therapy. Heterozygous loss of function mutations of three CCM genes (*KRIT1(CCM1)*, *CCM2*, and *PDCD10(CCM3)*) are associated with development of venous capillary dysplasias with hemorrhage and increased vascular permeability (*Mikati et al., 2015*) characteristic of CCM (*Leblanc et al., 2009*). Heterozygous patients often exhibit a 'second hit' on the normal *CCM* allele in CCM endothelial cells (*Akers et al., 2009*; *McDonald et al., 2011*) and loss of function of P53 or *Msh2*, genes that maintain genome stability, 'sensitize' *Krit1*[+/-] or *Pdcd10*[+/-] mice for development of CCM. Thus, CCMs are likely to arise following mosaic inactivation of both alleles of a given CCM gene. Neonatal endothelial-specific inactivation of murine *Krit1*, *Pdcd10 (Ccm3)*, or *Ccm2* results in cerebellar and retinal vascular lesions that resemble CCM (*Boulday et al., 2011*; *Chan et al., 2011*; *Jenny Zhou et al., 2016*). Thus, human CCMs arise from venous capillaries as a consequence of mosaic inactivation of these genes in endothelial cells.

The most striking form of CCM are large complexes containing adjacent dilated blood-filled thin-walled vessels with surrounding hemosiderin deposition indicative of chronic bleeding. These are the

lesions that are surgically resected either to relieve either mass effects or recurrent bleeding and are therefore clinically relevant. The cellular mechanism whereby these hallmark multi-cavernous lesions form is unknown and we reasoned that the genetic and experimental accessibility of the zebrafish and the optical transparency of its embryos and larvae (*Gore et al., 2018*) could enable in vivo analysis of CCM development. In particular, the fish is amenable to examination of genetic perturbations by study of mutant fish, CRISPR/Cas9 mutagenesis, or morpholino silencing (*Stainier et al., 2017*). The zebrafish 'heart of glass' defect in cardiac development led to the discovery of HEG1 (a binding partner for KRIT1; *Gingras et al., 2012*) and to the demonstration that loss of KRIT1 (*santa*) or its binding partner, CCM2 (*valentine*), produced an identical cardiac phenotype (*Mably et al., 2006*). Importantly, neither mutation nor silencing of any of these genes have been reported to produce zebrafish CCM. Second, silencing of *pdcd10* does not produce either cardiac dilation or CCM in the fish (*Yoruk et al., 2012*). Furthermore, there is compelling evidence that HEG1 mutations do not produce CCM in mice or humans (*Zheng et al., 2014*). These data indicate important differences between the consequences of deletion of these genes in the endocardium and in brain endothelial cells and underscore the need for a zebrafish model of authentic CCM.

Here, we used CRISPR-Cas9 mutagenesis of *ccm2* to recapitulate the mosaic inactivation of a CCM gene. We observed a highly penetrant novel phenotype in embryos. By 2 days post fertilization (dpf), ~30% of embryos developed striking segmental dilatation of the caudal vein associated with slowed blood flow and formation of multiple dilated contiguous blood-filled chambers. We show that these lesions are caused by the intraluminal extension of pillars that lead to physical obstruction of blood flow. These pillars resemble the first steps of intussusceptive angiogenesis; however, in contrast to the physiological process, these pillars fail to organize and fuse together to split the preexisting vessel in two. We show that this process of abortive intussusceptive angiogenesis is due to mosaic inactivation of *ccm2* leading to aberrant flow signaling and that, as in CCMs, KLF2 transcription factors play an important role in the formation of these lesions. In addition, typical CCM formed in the brains of all surviving adult and juvenile fish. As in murine CCM models, KLF2 transcription factors played a key role in the pathogenesis of zebrafish CCMs. Thus, abortive intussusceptive angiogenesis, as a consequence of aberrant flow signaling, leads to formation of multi-cavernous venous lesions in zebrafish embryos that resemble human multi-cavernous CCM.

## Results

### *ccm2* CRISPR zebrafish embryos display segmental dilation of the caudal venous plexus

As noted above, although null mutations of *krit1* and *ccm2* result in cardiac dilation and some vascular abnormalities, CCMs have not been observed in zebrafish (*Mably et al., 2006*; *Renz et al., 2015*). We reasoned because humans with CCM are mosaic for homozygous inactivation of *CCM1* or *CCM2*, that induction of such mosaicism using a CRISPR-Cas9 system (*Ablain et al., 2015*) could result in lesion formation. To create a mosaic animal, we co-injected Cas9 mRNA and gRNAs targeting the *ccm2* gene in zebrafish embryos (*Figure 1—figure supplement 1A*, *Supplementary file 2*). As expected, both genomic DNA sequencing and whole mount in situ hybridization showed that *ccm2* was targeted in a variable mosaic pattern affecting all tissues (*Figure 1—figure supplement 1B and C*). About half of these mosaic embryos displayed lethal cardiovascular defects.

The most prevalent lethal phenotype, observed in ~30% of 2 dpf *ccm2* CRISPR embryos, was localized dilatation of the caudal venous plexus (CVP) associated with erythrocyte accumulation and sluggish blood flow (*Figure 1A and B*, *Video 1*). This phenotype was clearly demonstrated in *Tg(fli1: EGFP)$^{y1}$* and *Tg(gata1:DsRed)$^{sd2}$* embryos in which erythrocytes are labeled with DsRed and endothelial cells with EGFP (*Figure 1C and D*). Examination of the dilated CVP revealed multiple large blood-filled chambers separated by thin-walled partitions that bore a resemblance to Stage 2 human multi-cavernous CCM (*Figure 1C*) in contrast to the normal architecture of control embryos (*Figure 1D*). In addition, ~5% of *ccm2* CRISPR embryos also displayed dilated cranial vessels (CV) (*Figure 1E*). We also noted expected phenotypes previously reported in *ccm2* morphants and mutants (*Mably et al., 2006*; *Renz et al., 2015*), a small proportion (~10%) of *ccm2* CRISPR embryos exhibited both heart dilation at 2 dpf and increased branching of the subintestinal vein (SIV) at 3 dpf (*Figure 1—figure supplement 2*). Co-administration of *ccm2* mRNA prevented both CVP dilation

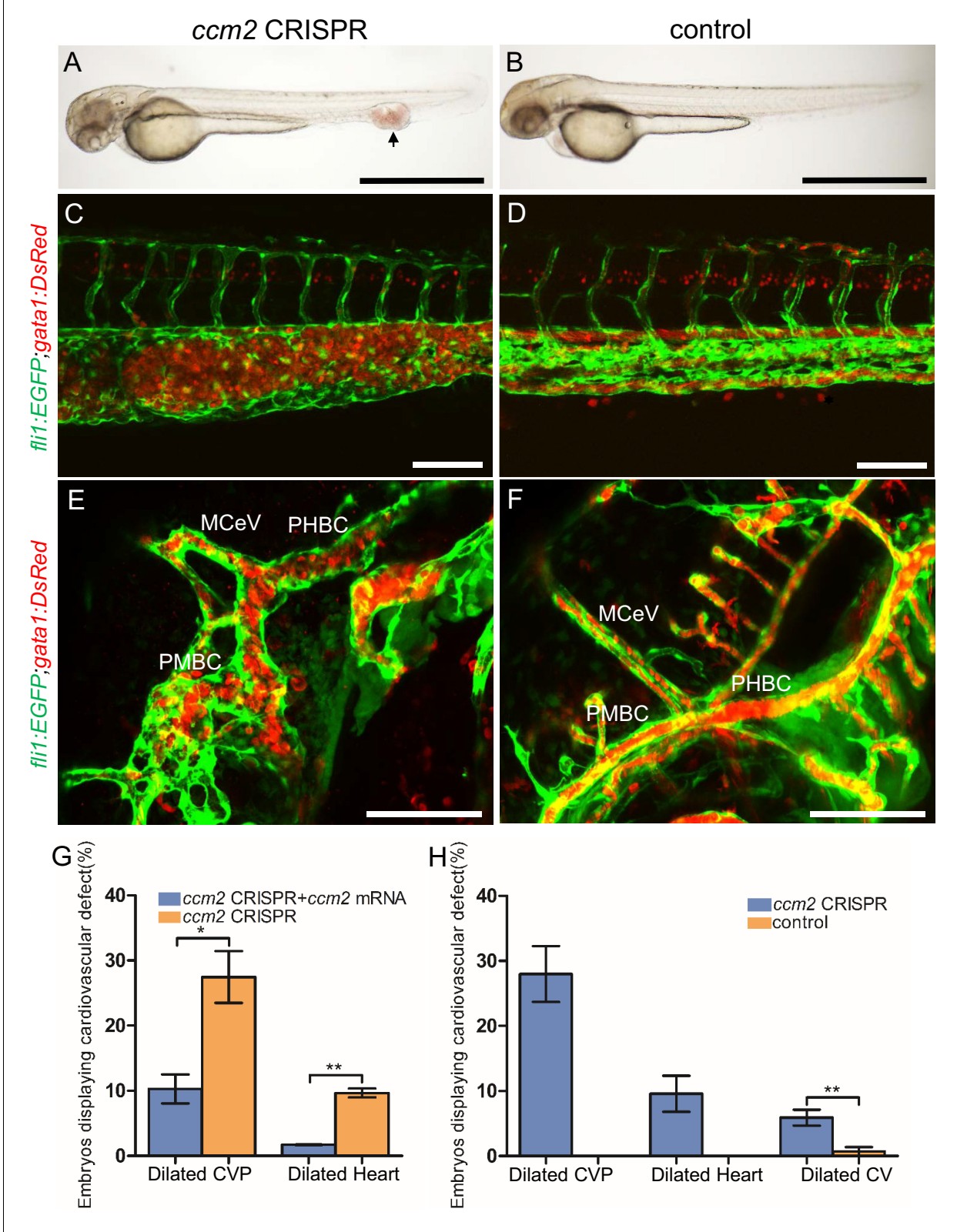

**Figure 1.** *ccm2* CRISPR zebrafish embryo display novel vascular phenotypes. Endothelial cells and red blood cells were labeled by EGFP and DsRed respectively in double transgenic *Tg(fli1:EGFP)$^{y1}$;Tg(gata1:DsRed)$^{sd2}$* embryos. (**A**) Red blood cells accumulate in dilated segments of the caudal vein of *ccm2* CRISPR fish at 2 days post fertilization (dpf). (**B**) cas9 mRNA-injected control embryo. (**C**) *ccm2* CRISPR embryos showed accumulation of red blood cells and intraluminal endothelial cells in a dilated segment of caudal vein in contrast to a control embryo. Note: In this and all succeeding

*Figure 1 continued on next page*

*Figure 1 continued*

sagittal views, anterior is to the left (D). (E) *ccm2* CRISPR embryos occasionally showed dilations of cerebral veins, whereas control embryos (F) showed normal development of cerebral veins (F). MCeV: mid-cerebral vein, PMBC: primordial midbrain channel, PHBC: primordial hindbrain channel. (G) The dilated caudal venous plexus (CVP) and heart of *ccm2* CRISPR embryos were rescued by ccm2 mRNA injection. p=0.0336 (dilated CVP), 0.0037 (dilated heart). p-Values were calculated using an unpaired two-tailed Student's t-test. (H) Phenotypic distribution of dilated heart, CVP, and cerebral veins (CV) in *ccm2* CRISPR embryos at 2 dpf. p=0.0078 (dilated CVP), 0.0268 (dilated heart), 0.0041 (dilated CV). p-Values were calculated using a paired two-tailed Student's t-test. Error bars indicate SD. Scale bar: 1 mm in A and B, and 100 μm in C through F.

The online version of this article includes the following figure supplement(s) for figure 1:

**Figure supplement 1.** *ccm2* CRISPR zebrafish exhibit mosaic expression of CCM2.

**Figure supplement 2.** Approximately 10% of *ccm2* CRISPR zebrafish embryos exhibited dilation of heart and increased branch points of subintestinal vein.

and heart dilation in *ccm2* CRISPR embryos (*Figure 1G*) confirming that both phenotypes are due to *ccm2* loss. Notably, the dilated heart and CVP dilation appeared to be mutually exclusive, that is, in over 200 embryos analyzed, we never observed both phenotypes in a single *ccm2* CRISPR embryo. Thus, the localized CVP dilation was the most prevalent phenotype observed in 2 dpf *ccm2* CRISPR embryos in comparison to the cardiac and SIV phenotypes that characterize *ccm2* null and *ccm2* morphant embryos (*Figure 1H*; *Mably et al., 2006*; *Renz et al., 2015*).

## Abortive intussusceptive angiogenesis in the dilated CVP of ccm2 CRISPR embryos

We used confocal microscopy and three-dimensional (3D) reconstruction of the dilated area of the CVP in *Tg(fli1:EGFP;gata1:DsRed)* zebrafish to explore their underlying structural defect. We noted intraluminal endothelial pillars that partitioned the lumen (*Figure 2A* through C) of the dilated CVP. In contrast, as expected, a completely patent caudal and ventral vein lumen formed in control embryos (*Figure 2D* through F). Furthermore, the ventral vein, which normally forms by a combination of sprouting and intussusceptive angiogenesis (*Karthik et al., 2018*), was lost in the dilated area of the CVP (*Figure 2A* through F). Importantly, examination of 3D reconstructions of the vessel revealed that these pillars were associated with pits on the external surface of the dilated CVP (*Figure 2G* arrows), a hallmark of the initial phase of intussusceptive angiogenesis (*Djonov et al., 2003*). In contrast to normal intussusceptive angiogenesis, wherein transluminal pillars ultimately fuse to divide vessels longitudinally into new daughter vessels (*Djonov et al., 2003*), the intussusceptions observed in *ccm2* CRISPR embryos were not coordinately formed and failed to fuse resulting in a honeycombed lumen. This honeycombing created a lumen with multiple chambers filled with red blood cells (RBCs) associated with sluggish blood flow (*Figure 2I* through K, *Videos 2* and *3*), whereas patent lumens and normal blood flow were observed in control embryos (*Video 4*). Thus, in these mosaic *ccm2* null zebrafish, an expanded region of the CVP is formed by multiple dilated erythrocyte-filled chambers and is associated with evidence of incomplete intussusceptive angiogenesis.

## Ablation of intravascular pillars reverts the dilated CVP phenotype

The multiple dilated compartments and sluggish blood flow suggested that this phenotype could be due to obstruction of free flow of erythrocytes by the meshwork of intravascular pillars. In support of this idea, we observed that spontaneous regression of an existing pillar was accompanied by reduced dilation of the CVP (*Figure 3A* through D). In addition, the trapped erythrocytes began to circulate freely. This rapid relief of both vessel dilation and blood stagnation suggested

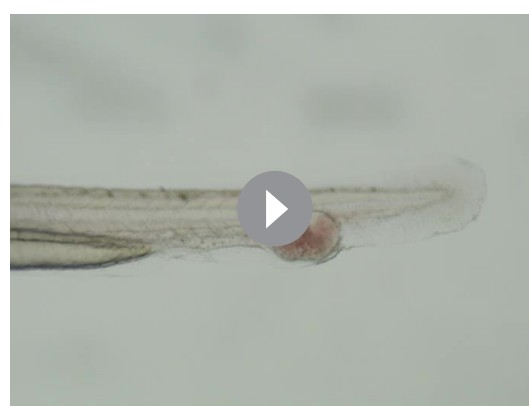

**Video 1.** On 2 days post fertilization (dpf), *ccm2* CRISPR embryo displayed cavernoma-like lesion in the tail. Blood flow was slowed down in the lesion area that contained retained blood cells.

https://elifesciences.org/articles/62155#video1

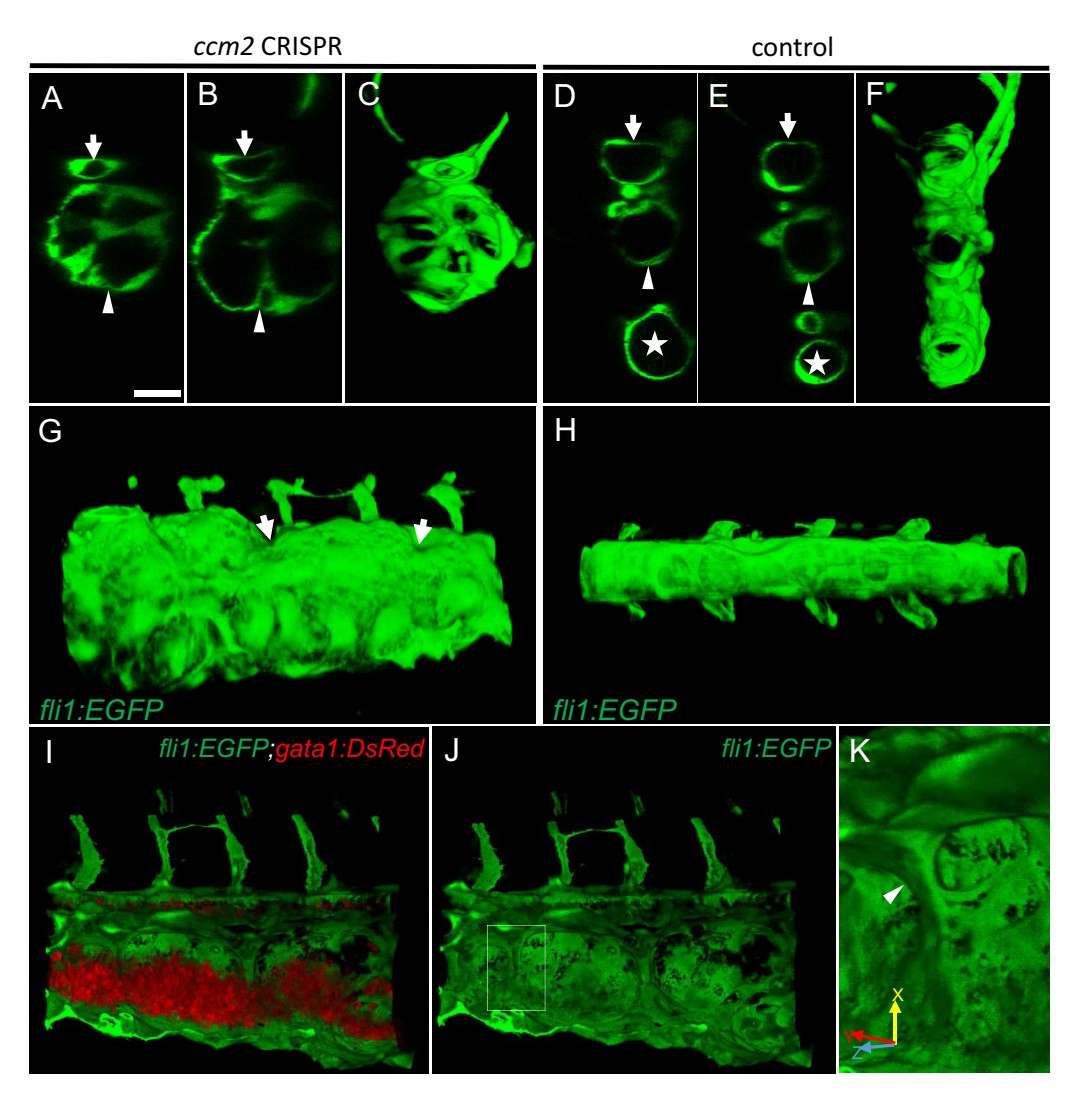

**Figure 2.** Intravascular pillars honeycomb the lumen of the caudal vein in *ccm2* CRISPR embryos. (**A–F**) XZ planes and three-dimensional (3D) projection along Y axis of Airyscan images revealed intraluminal endothelial pillars at 2 days post fertilization (dpf) (**A–C**), whereas Cas9-injected control embryos displayed a normal patent lumen in both a dorsal and ventral caudal vein (**D–F**). Endothelial cells were labeled by EGFP in *Tg(fli1:EGFP)* embryos. Arrow, arrowhead, and asterisk indicated the dorsal aorta, dorsal vein, and ventral vein, respectively. (**G and H**) Ventral view of 3D reconstruction show the irregular surface of the dramatically dilated caudal vein segment in *ccm2* CRISPR embryo (**G**) and normal ventral vein (**H**). Arrows in G indicate small pits where the endothelial pillars originate. (**I–K**) Intraluminal view of 3D reconstruction of *ccm2* CRISPR embryo reveals the intraluminal pillars honeycombing the lumen and the accumulated red blood cells (**I**). Erythrocytes were not imaged in J to reveal pillars and the area within the box in (**J**) was magnified in (**K**), and arrowhead indicates the intravascular pillar. Endothelial cells and red blood cells were labeled by EGFP or DsRed respectively in *Tg(fli1:EGFP)^{y1};Tg(gata1:DsRed)^{sd2}* embryos. Scale bar: 20 µm.

that the aberrant pillars may form a physical barrier thus resulting in accumulation of erythrocytes in dilated cavernous structures. To directly test the role of obstruction by intravascular pillars in dilation, we used targeted short pulses of near-infrared laser light to sever the pillars, a technique that generates negligible heat transfer and collateral damage to neighboring tissues (*Nishimura et al., 2006*). There was near instantaneous reduction of the dilated vessel diameter (93.4 µm) to near-normal dimensions (69.4 µm) in the example shown (*Figure 3E and F*, *Video 5*). In three such independent experiments, severing these pillars resulted in a 29 ± 4% reduction in vessel diameter (p=0.0004, two-tailed t-test). Thus, the pillars are an underlying cause of the CVP dilation observed in *ccm2* CRISPR embryos.

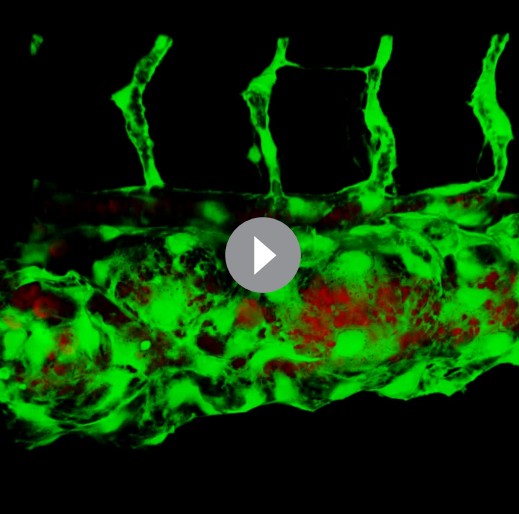

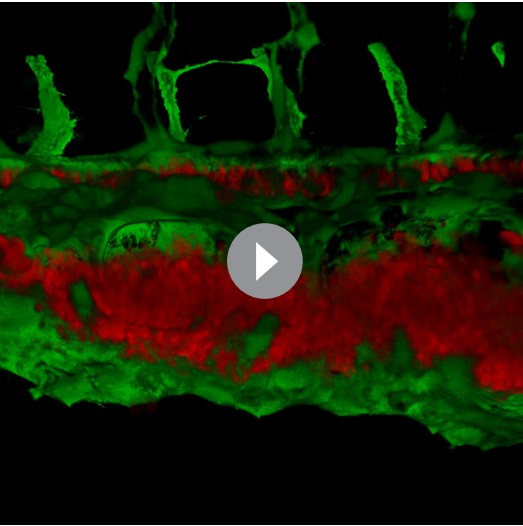

**Video 2.** Three-dimensional exterior view of caudal venous plexus (CVP) of 2 days post fertilization (dpf) *ccm2* CRISPR embryo. Note pits on the surface and that the CVP is partitioned into several dilated areas.
https://elifesciences.org/articles/62155#video2

**Video 3.** Three-dimensional interior view of caudal venous plexus (CVP) of 2 days post fertilization (dpf) *ccm2* CRISPR embryo. Note the endothelial pillars within the lumen and accumulated red blood cells.
https://elifesciences.org/articles/62155#video3

## Blood flow and red blood cells are required for CVP dilation

The importance of the pillars in CVP dilation suggested that obstruction of blood flow was responsible for the phenotype. Consistently, as noted above, *ccm2* CRISPR embryos displaying CVP dilation did not show heart dilation. Conversely, segmental CVP dilation was absent in *ccm2* null mutants or *ccm2* morphants that exhibit characteristic heart dilation (*Figure 3—figure supplement 1*). These observations suggest that a normally pumping heart and thus normal blood flow is required for CVP dilation. To investigate the role of blood flow, we took advantage of the capacity of zebrafish embryos to obtain sufficient oxygen by diffusion to survive temporarily in the absence of circulating blood. We induced a silent heart phenotype by

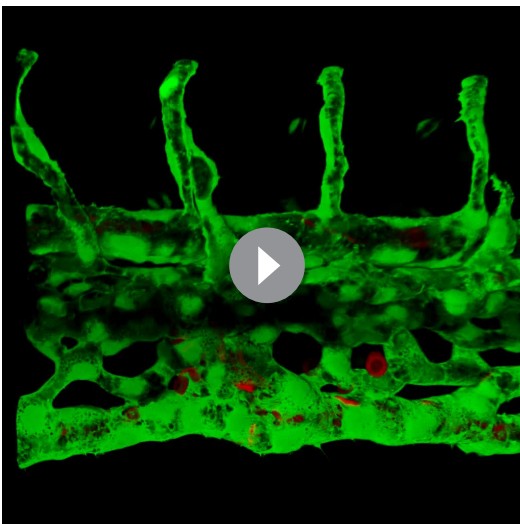

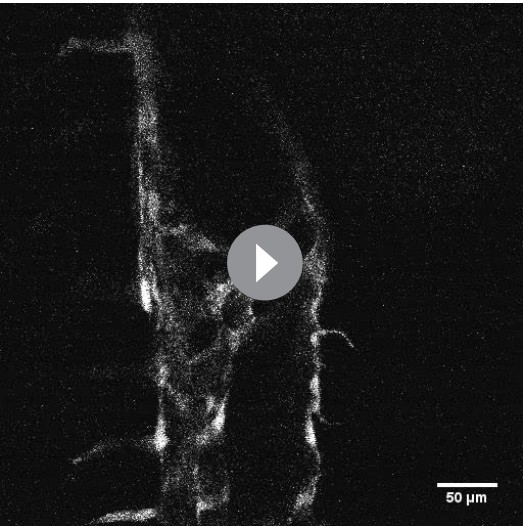

**Video 4.** Three-dimensional view of caudal venous plexus (CVP) of 2 days post fertilization (dpf) control embryo.
https://elifesciences.org/articles/62155#video4

**Video 5.** Laser ablation of intussusception reduced the vessel diameter.
https://elifesciences.org/articles/62155#video5

using a troponin T (*tnnt*) morpholino, resulting in ~65% reduction in the frequency of CVP dilation (*Figure 4A*). We also reasoned that the meshwork of pillars would not obstruct fluid flow but would present a barrier to free passage of erythrocytes. Reduction of erythrocytes using morpholinos directed against *gata1*(*Galloway et al., 2005*) or *tif1-γ* (*Monteiro et al., 2011*) transcription factors produced a similar dramatic reduction in the CVP dilation (*Figure 4A*). These data indicate that the meshwork of pillars obstructs the passage of erythrocytes in flowing blood resulting in multiple erythrocyte-filled cavernous chambers that dilate the CVP.

As shown in *Figure 2*, the sprouts that form the ventral vein are lost in the dilated region of the CVP. Because CVP development in *tnnt* morphants is nearly normal (*Choi et al., 2011*), we inspected the regions of the CVP displaying loss of ventral sprouting in *tnnt* morphant *ccm2* CRISPR embryos. In 11 such embryos, in spite of the defective ventral sprouting and ventral vein formation, we observed no intravascular pillars. This result suggests that blood flow, in addition to causing the CVP dilation, is required for intussusceptive pillar formation (*Figure 4B and C*) as it is for normal CVP arborization (*Karthik et al., 2018*).

Inactivation of either *ccm1* or *ccm2* markedly upregulates expression of KLF2, a flow-regulated transcription factor required for normal cardiovascular development and for CCM formation (*Renz et al., 2015*; *Zhou et al., 2015*; *Zhou et al., 2016*). In situ hybridization revealed that *klf2a* was also upregulated in the CVP of *ccm2* CRISPR embryos (*Figure 4—figure supplement 1*). We used a *klf2a* reporter line, *Tg(klf2a:H2AEGFP)*, together with an endothelial cell-specific marker line (*Tg(kdrl:mcherry)^is5*) to observe the activity of the *klf2a* promoter. *Ccm2* morphants displayed a generalized increase in *klf2a* reporter expression (*Figure 4—figure supplement 2A and A'*), whereas the absence of blood flow in the *tnnt* morphant caused much reduced reporter expression in endothelial cells (*Figure 4—figure supplement 2B and B'*; *Figure 4—figure supplement 2C and C'*). Consistent with previous reports (*Parmar, 2006*; *Renz et al., 2015*), these opposing changes confirm that *ccm2* and flow can regulate expression of KLF2a. In 23 hpf *ccm2* CRISPR embryos, examined prior to onset of blood flow, a patchy increase in *klf2a* reporter expression was observed in *ccm2* CRISPR endothelial cells (*Figure 4D*), whereas reporter expression was uniformly low in control embryos at the same stage (*Figure 4E*). A quantitative analysis revealed a subpopulation of high KLF2a-expressing endothelial cells in *ccm2* CRISPR embryos that was absent in control embryos (*Figure 4F*). Furthermore, in *tnnt* morphant 2 dpf *ccm2* CRISPR embryos, there was also a striking mosaic increase in endothelial *klf2a* reporter expression (*Figure 4G*). Thus, dilation was associated with the patchy upregulation of a flow-sensitive transcription factor, KLF2, in the *ccm2* CRISPR CVP. Taken together, these results suggest that patchy KLF2 expression in combination with blood flow leads to formation of these dilated RBC-filled multi-cavernous lesions in the CVP.

## Mosaic upregulation of KLF2a is sufficient for cavernoma formation in CVP

The patchy increase in KLF2a expression in the CVP endothelial cells of *ccm2* CRISPR embryos and requirement for blood flow suggested the possibility that these two factors led to the formation of cavernomas in the CVP. To address the role of KLF2, we injected *klf2a* and *klf2b* morpholinos and observed reversal of both CVP dilation and heart dilation in the *ccm2* CRISPR embryos (*Figure 5A*). Furthermore, *ccm2* CRISPR treatment of *klf2a^-/-* embryos caused no CVP dilation (*Figure 5—figure supplement 1*). Thus, *klf2a* is required for the CVP dilation phenotype.

In *ccm2* CRISPR embryos, KLF2a was both upregulated in a mosaic fashion and required for CVP dilation; we therefore asked whether mosaic upregulation of KLF2a expression per se causes cavernoma formation. Mosaic overexpression was accomplished by injecting a plasmid encoding KLF2a into *Tg(fli1:EGFP)^y1* embryos; ~6% of such embryos displayed CVP dilation compared to control embryos injected with ΔKLF2a plasmid expressing KLF2a with a deleted DNA binding domain (*Oates et al., 2001*; *Figure 5B and C*). Affected embryos exhibited intussusceptions within the CVP lumen accompanied by dilation (*Figure 5D*). These observations show that mosaic upregulation of KLF2a expression is sufficient for cavernoma formation when blood is flowing.

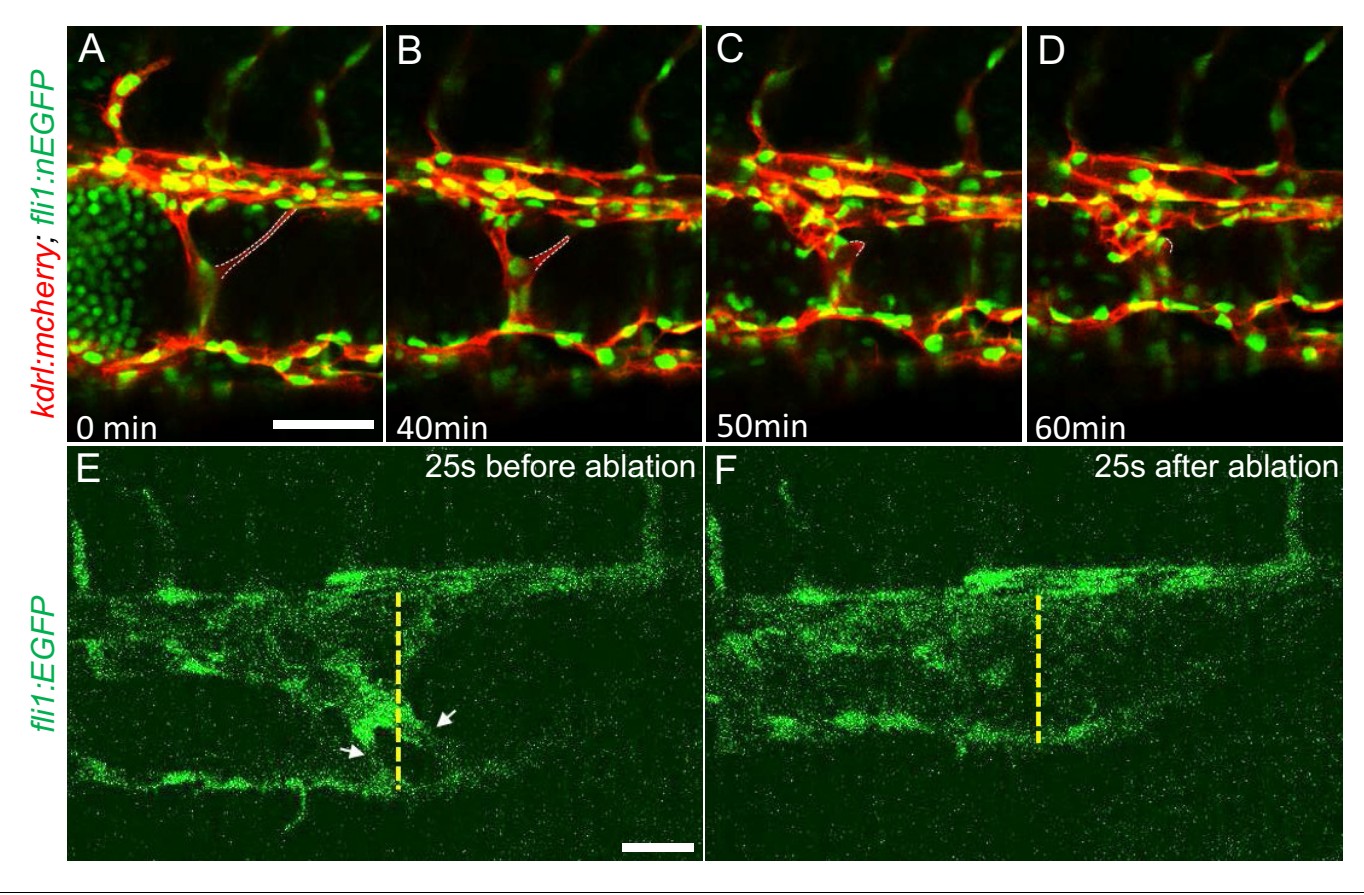

**Figure 3.** Intravascular pillars obstruct blood flow leading to vessel dilation in *ccm2* CRISPR embryos. (**A** through **D**) Time lapse images reveal spontaneous retraction of an intravascular pillar leading to re-entry of blood cells into circulation and reduced dilation of the caudal vein. Endothelial cells were labeled by mCherry, and their nucleus and some red blood cells were labeled by EGFP in the *Tg(fli1:nEGFP)^{y7};Tg(kdrl:mcherryras)^{s896}* embryos. The retracted pillar is outlined by dotted lines for emphasis. Note that pillar retraction and vessel dilation were temporally correlated. (**E** and **F**) Laser ablation of pillar reduced caudal venous plexus (CVP) diameter. The diameter of the dilated vein (**E**) was reduced after ablation (**F**). Note the pillars indicated by arrows in (**E**) are gone after ablation in (**F**). Dashed line indicates the diameter of the vein before and after ablation. Scale bar: 50 μm.

The online version of this article includes the following figure supplement(s) for figure 3:

**Figure supplement 1.** Both *ccm2* null mutants and morphants displayed heart dilation but no caudal venous plexus (CVP) dilation on 2 days post fertilization (dpf).

## Mosaic expression of ccm2 causes KLF2a-dependent cavernoma formation

*ccm2* CRISPR caused mosaic inactivation of *ccm2* and the dilated CVP phenotype, whereas global inactivation of *ccm2* in *ccm2* null mutants or *ccm2* morphants does not. We therefore questioned whether mosaicism, per se, played a role in the CVP dilation. To test this idea, we globally reduced *ccm2* expression by co-injecting a sublethal dose of *ccm2* morpholino with the *ccm2* gRNA CRISPR mixture. The chosen morpholino dose did not increase the frequency of observable heart defects; however, the percentage of embryos displaying CVP dilation decreased dramatically (*Figure 6A*). We then reasoned that because *ccm2* acts as a scaffold connecting *krit1* to *ccm3* (*Stahl et al., 2008*), the overexpression of *ccm2* might have a dominant negative effect. Indeed, when we injected linearized DNA containing *ccm2* fused to m-Orange, *ccm2* mosaic overexpression led to CVP dilation and aberrant intussusceptions similar to those observed in *ccm2* CRISPR embryos in ~8% of embryos (*Figure 6B and B'*). In sharp contrast, injection of a plasmid encoding a loss of krit1 binding function *ccm2(L197R)* mutant (*Kleaveland et al., 2009*) resulted in of embryos displaying a normal vascular development (*Figure 6C and C'*). Importantly, mosaic overexpression of *ccm2* caused

significantly less CVP dilation in *klf2a*$^{-/-}$ embryos (*Figure 6—figure supplement 1*). Thus, mosaicism for *ccm2* expression causes *klf2a*-dependent formation of multi-cavernous erythrocyte-filled structures in the CVP. Combined with the capacity of mosaic expression of *klf2a* to cause CVP dilation, these results show that mosaic expression of CCM2 leads to mosaic KLF2a expression and abortive intussusceptive angiogenesis that obstructs the lumen to form these cavernoma-like lesions.

## CCMs in adult zebrafish

The foregoing data indicated that mosaic inactivation of *ccm2* results in a multi-cavernous lesion in the embryonic CVP that resembles mammalian CCM in gross architecture and dependence on KLF2. We then asked if authentic CCM would develop in the ~50% of *ccm2* CRISPR embryos that developed with a normal gross morphology and survived to adulthood. Brain vascular lesions were observed in virtually all of these adult *ccm2* CRISPR zebrafish (*Figure 7A and C*) and not in control fish (*Figure 7E and G*). In order to image the lesions at the whole brain level, clear, unobstructed brain imaging cocktails and computational analysis (CUBIC) was applied to these brains, and the transparent brains were scanned by light sheet microscopy (*Figure 7B,D,F and H*). The distribution of lesions included cerebrum, cerebellum, brain stem, and, in some fish, the spinal cord (*Figure 7I*). This distribution pattern is similar to that found in patients (*Goldstein and Solomon, 2017*). Hematoxylin and eosin stained sections showed dilated multi-cavernous vascular channels filled with nucleated erythrocytes and lacking mature vessel wall angioarchitecture (*Figure 7J*). Perl's Prussian blue staining indicated prior hemorrhage adjacent to the lesions (*Figure 7K*). These histological findings were absent in control fish (*Figure 7L and M*) and resemble those in CCM patients (*Figure 7N and O*; *Cox et al., 2017*). A dramatic reduction in CCM was seen in *ccm2* CRISPR in *klf2a*$^{-/-}$ zebrafish (*Steed et al., 2016*; *Figure 7P*) consistent with previous murine studies in which inactivation of *Klf2* prevented CCM formation (*Zhou et al., 2016*). In addition, similar to humans, these adult zebrafish also developed extracranial lesions (*Figure 7—figure supplement 1*).

## Discussion

Familial CCM lesions form as a consequence of mosaic complete inactivation of *CCM1, -2,* or *-3*. Here, we have used Cas9-CRISPR mutagenesis to create such a mosaicism for *ccm2* in zebrafish and show that surviving adult *ccm2* CRISPR animals develop brain and extracranial lesions that closely resemble those observed in humans with CCM. In ~30% of embryos, we observed a novel phenotype, the formation of segmental dilatation of the caudal vein associated with slowed blood flow and formation of multiple markedly dilated blood-filled chambers, resembling a multi-cavernous CCM. These lesions are caused by intussusceptive intraluminal pillars that obstruct the passage or erythrocytes resulting in the development of multiple dilated blood-filled chambers. These pillars form as a consequence of a combination of blood flow and mosaic overexpression of a flow-dependent transcription factor, KLF2a, leading to aberrant flow sensing in the developing CVP. In sum, our studies describe a zebrafish model for CCM and provide a new mechanism that can explain the formation of the characteristic multi-cavernous lesions seen in humans.

### The role of blood flow in CVP dilation

The segmental dilation of the CVP is due to intussusceptive pillars that fail to fuse normally, thus honeycombing the vein lumen and obstructing the free flow of erythrocytes. Evidence for the role of obstruction includes the marked slowing of blood flow within the lesions, dependence of dilation on blood flow and erythrocytes, and the relief of dilation by spontaneous or induced regression of the pillars. Intussusceptive angiogenesis differs from sprouting angiogenesis by splitting the existing vessel intraluminally as a response to increased blood flow (*Djonov et al., 2003*; *Egginton et al., 2001*). Recent elegant studies have shown that localized reduction in fluid shear stress occurs adjacent to intussusceptive pillars and is associated with the formation of new pillars that align with existing pillars (*Karthik et al., 2018*). These observations suggested that these blood flow patterns are responsible for the formation of the aligned pillars required for orderly fusion to split the vessel in two (*Karthik et al., 2018*). Similar to physiological intussusceptive angiogenesis, the hallmark pits and intraluminal pillars were also observed in the CVP of *ccm2* CRISPR embryos; however, these pillars failed to undergo orderly fusion to split the vessel. We propose that this failure to undergo orderly fusion and CVP arborization is due to mosaic overexpression of *klf2a*, a flow-sensitive

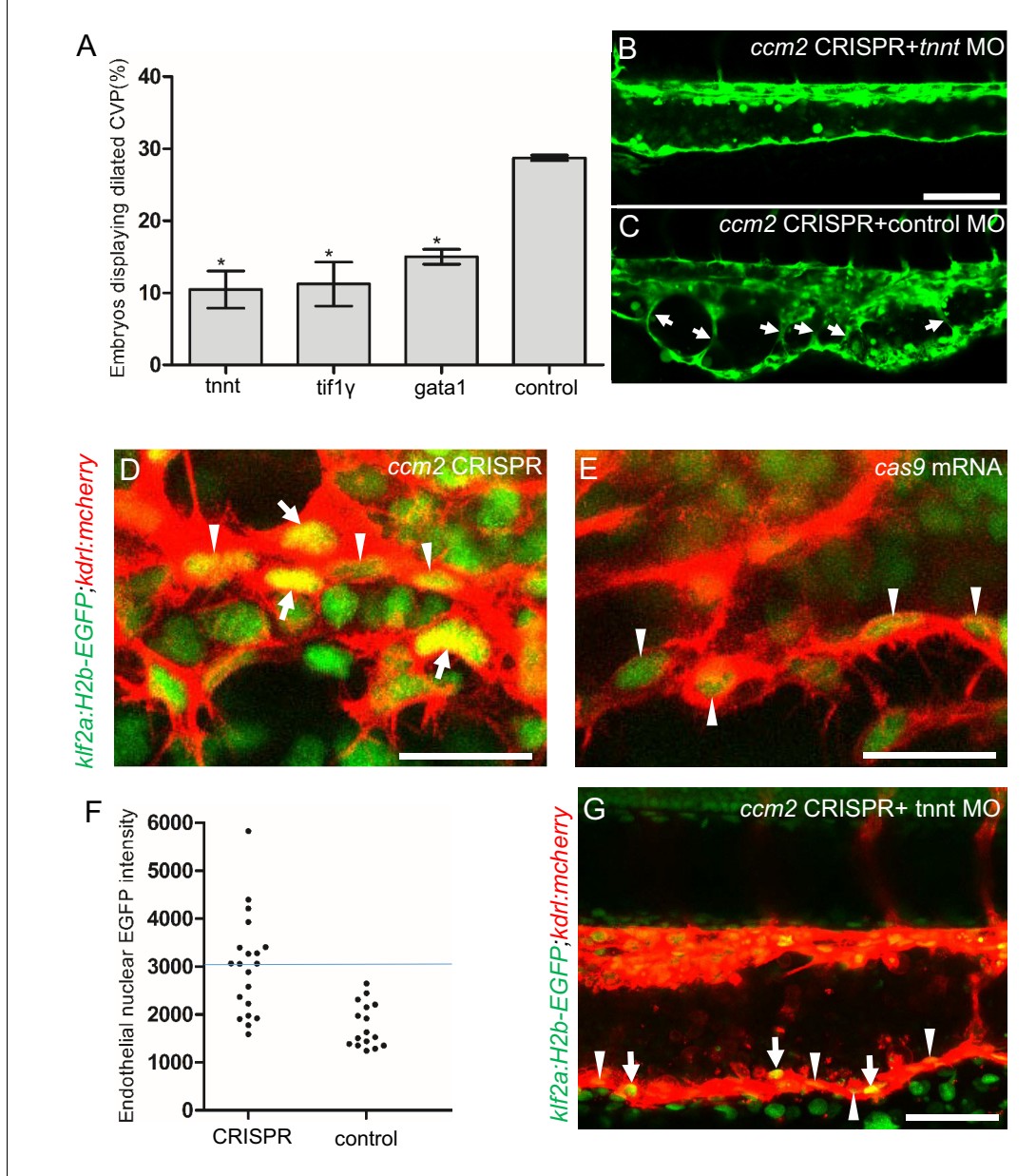

**Figure 4.** Blood flow is required for pillar formation and vessel dilation. Morpholinos targeting *tnnt*, *gata1*, *tif1gamma*, or a control morpholino were co-injected with *ccm2* guide and Cas9 RNA. (A) Reduction of blood flow in *tnnt* morphants (A, B, C) resulted in reduced caudal venous plexus (CVP) dilation (A) and intravascular honeycombing (B, C) in 2 days post fertilization (dpf) *ccm2* CRISPR *Tg(fli1:EGFP)* embryos. Arrows indicate intussusceptions. Scale bar: 100 μm. (A) Loss of erythrocytes in *gata1 or tif1gamma* morphant *ccm2* CRISPR embryos also reduced the incidence of CVP dilation. p-Values were calculated using one-way ANOVA. **p<0.01. Error bars indicate SD. (D and E) At 23 hpf, *ccm2* CRISPR *Tg(klf2a:H2b-EGFP)* embryos displayed a mosaic increase of EGFP expression in endothelial cells in the CVP (D), compared with cas9 mRNA control embryos (E). Scale bar: 25 μm. (F) Quantification of the EGFP fluorescence intensity using ImageJ. A total of 20 nuclei were analyzed from *ccm2* CRISPR embryos, and 16 nuclei were analyzed from control embryos. Note that 11 nuclei in CRISPR embryo displayed intensity above 3000, while all of the nuclei in control embryo are below 3000. (G) *ccm2* CRISPR and *tnnt* morpholino-injected *Tg(klf2a:H2b:EGFP* 2 dpf) embryos displayed a mosaic increase of endothelial nuclear EGFP expression in dorsal vein. Scale bar: 50 μm. In A through C, EGFP expression was driven by *klf2a* promoter in *Tg(klf2a:H2b:EGFP)* embryo, and endothelial cells were labeled by mcherry in *Tg(kdrl:mcherry)* transgenic line. Arrows indicated the endothelial nuclei with increased EGFP, and arrowheads indicated the other endothelial nuclei along the ventral wall of dorsal vein.

The online version of this article includes the following figure supplement(s) for figure 4:

**Figure supplement 1.** Whole mount in situ hybridization showed mosaic upregulation of klf2a expression in *ccm2* CRISPR embryos (left) compared to that of control (right).

**Figure supplement 2.** Klf2a expression is regulated by ccm2 expression and by blood flow.

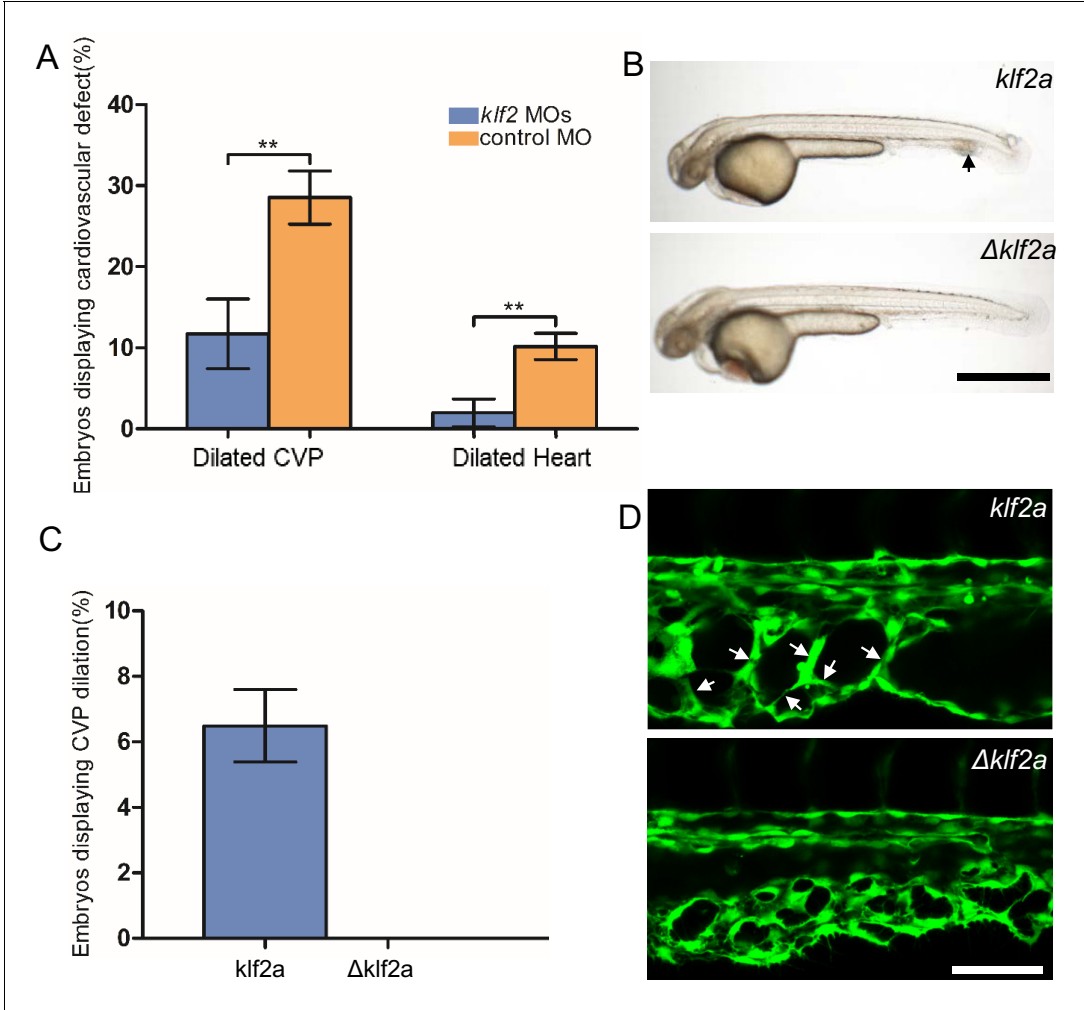

**Figure 5.** Mosaic KLF2a expression caused caudal venous plexus (CVP) dilation. (**A**) Both the CVP dilation and heart dilation were rescued by injection of *klf2* morpholinos in 2 days post fertilization (dpf) *ccm2* CRISPR embryos. \*\*p<0.01. Error bars indicate SD. (**B**) *pCS2*-KLF2a linearized DNA-injected 2.5 dpf embryos displayed CVP dilation, whereas injection of a DNA fragment containing a DNA binding domain deleted ΔKLF2a mutant showed normal development. Arrow indicates the CVP dilation and retained erythrocytes. Scale bar: 1 mm. (**C**) Quantification of the prevalence of CVP dilation following KLF2a or ΔKLF2a overexpression. The mean and SD are shown. (**D**) Representative images show the honeycombed lumen and dilated CVP in 1.5 dpf KLF2a-injected embryo and normal CVP of ΔKLF2a-injected embryo. Arrow indicates honeycombing. Scale bar: 100 μm.

The online version of this article includes the following figure supplement(s) for figure 5:

**Figure supplement 1.** Reduced caudal venous plexus (CVP) dilation in *ccm2* CRISPR *klf2a*⁻/⁻ embryos.

transcription factor, thus disrupting the required orderly flow signaling (*Karthik et al., 2018*). The meshwork formed by these intraluminal endothelial pillars partitions the patent lumen into multiple blood-filled chambers (*Figure 3I and J*, *Videos 2* and *3*). As more and more RBCs accumulate, the CVP becomes dilated (*Video 1*).

In contrast to the necessity of blood flow for formation of the multi-cavernous CVP lesions, in *ccm2* CRISPR zebrafish, blood flow suppresses endothelial proliferation and simple vessel dilation in *krit1* global null zebrafish (*Rödel et al., 2019*). As shown here, vessels mosaic for expression of a CCM gene require blood flow to form the intravascular pillars that obstruct blood flow and cause multi-cavernous lesions. Previous studies termed dilated capillaries Stage 1 CCM and multi-cavernous lesions Stage 2 CCM (*Zeineddine et al., 2019*). The differential flow requirements for formation of dilated vessels and multi-cavernous lesions in zebrafish suggest that the Stage 1 and Stage 2 forms of CCM can employ distinct pathogenetic mechanisms.

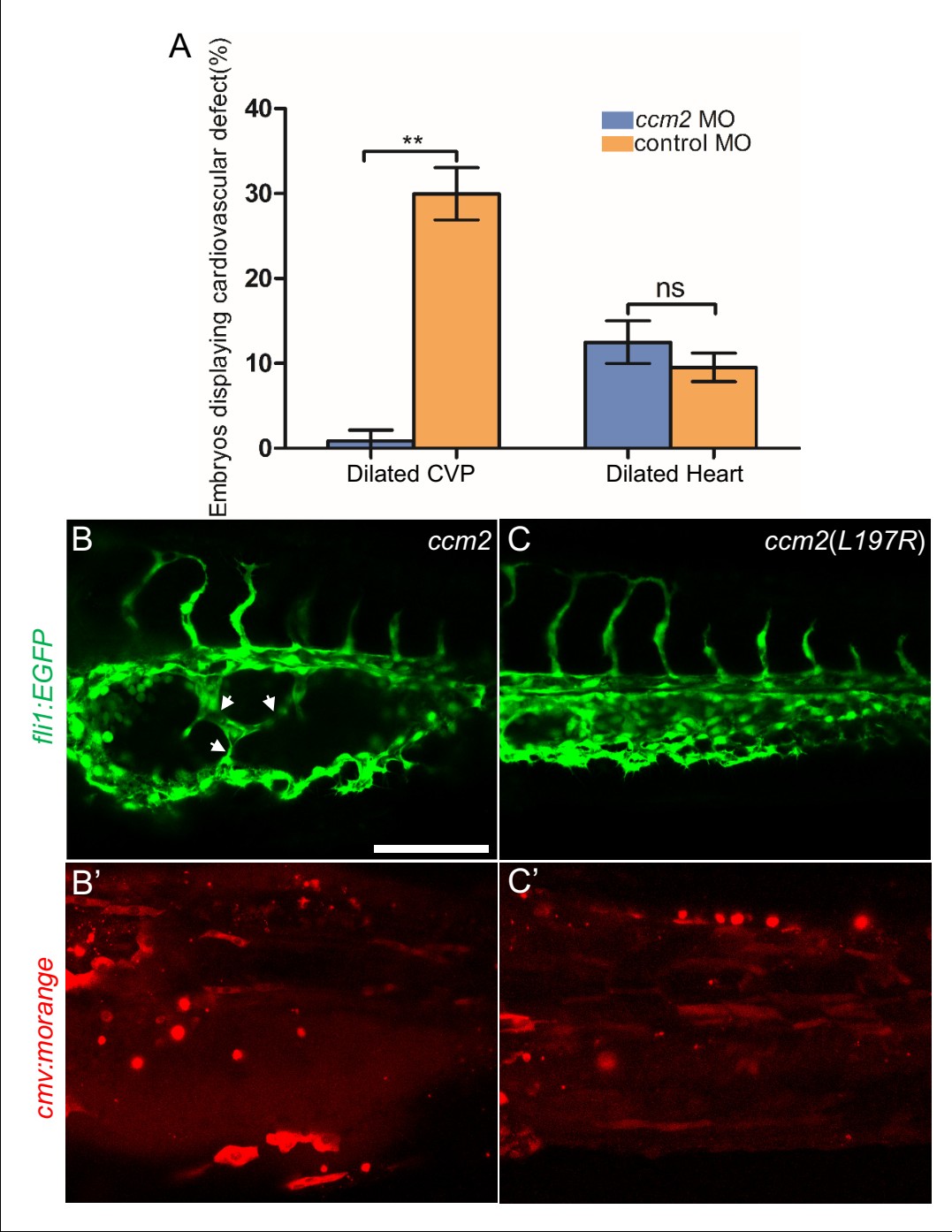

**Figure 6.** Mosaic *ccm2* expression caused caudal venous plexus (CVP) dilation. (**A**) Low-dose *ccm2* morpholino reduced the incidence of CVP dilation but did not significantly increase heart dilation in *ccm2* CRISPR embryos. (**B** and **C**) Mosaic *ccm2* but not inactive *ccm2(L197E)* overexpression caused CVP dilation. Arrows indicate pillars in the CVP. (**B'** and **C'**) Mosaic expression of mOrange-tagged *ccm2* or *ccm2(L197E)*. Scale bar: 100 μm. Error bars are ± SD.

The online version of this article includes the following figure supplement(s) for figure 6:

**Figure supplement 1.** Reduced caudal venous plexus (CVP) dilation in CCM2 over expressing *klf2a⁻/⁻* embryos.

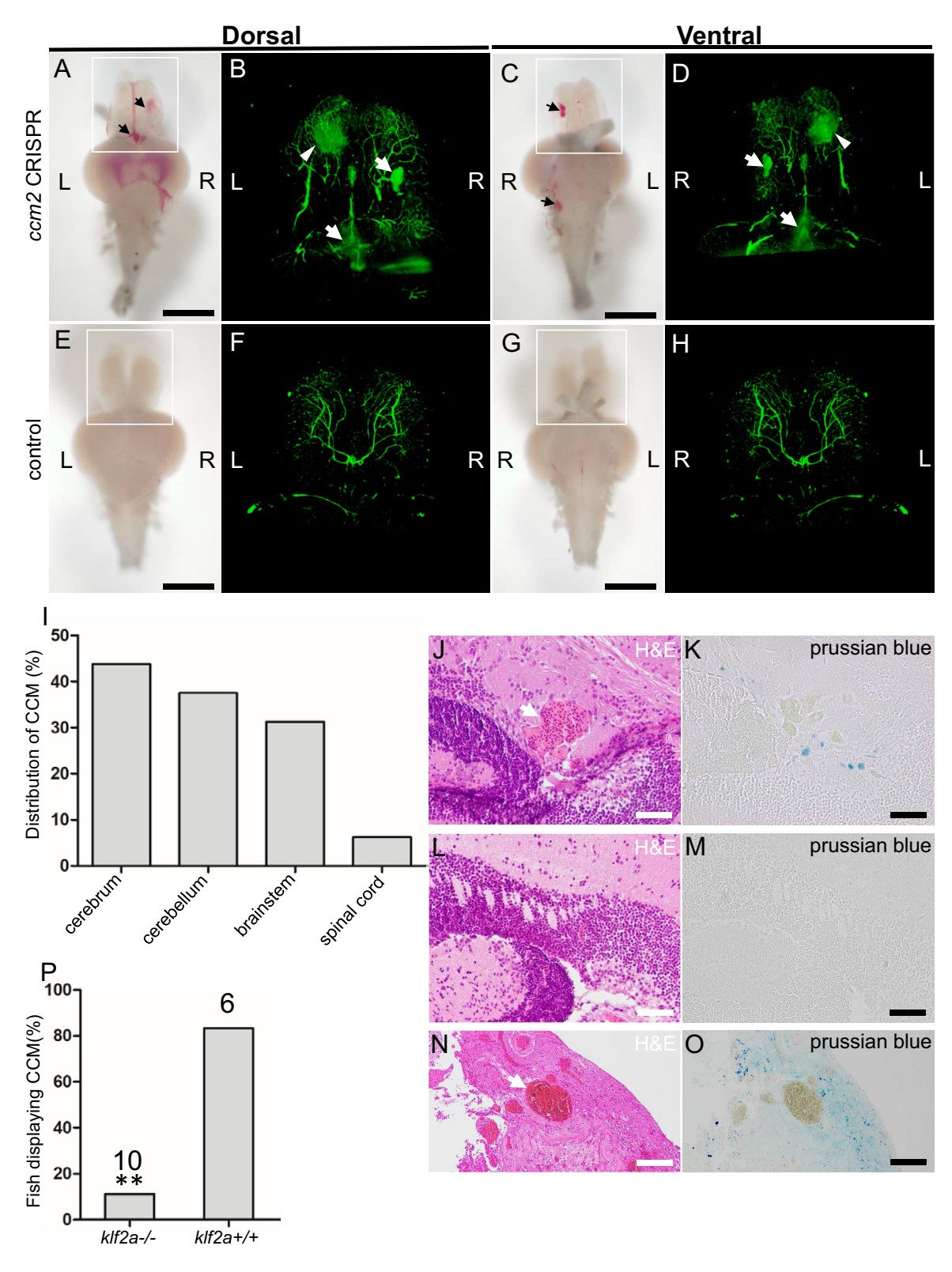

**Figure 7.** Adult *ccm2* CRISPR zebrafish develop typical cerebral cavernous malformation (CCM) lesions. The ~50% of *ccm2* CRISPR fish that survived developed highly penetrant CCMs (**A and C**). Arrows indicate superficial lesions on dorsal (**A**) and ventral (**C**) surface of the brain. Note hemorrhage into the ventricles. Lesions are absent in control embryos (**E and G**). Clear, unobstructed brain imaging cocktails and computational analysis (CUBIC) clearing (**B, D, F, H**) enables visualization of CCM burden by light sheet microscopy. Arrows indicate the lesions that corresponded to those seen in
*Figure 7 continued on next page*

*Figure 7 continued*

bright field, and arrowhead indicates a deeper lesion. L: left, R: right. Scale bar: 1 mm. (I) Cavernomas were dispersed throughout the central nervous system including cerebrum, cerebellum, brain stem, and spinal cord. (J) Hematoxylin and eosin (H&E) stained brain section reveals nucleated erythrocytes filling a dilated vessel with adjacent Prussian blue stained iron deposition (K) in *ccm2* CRISPR fish and the absence of lesions or iron deposition in control fish (L, M). (N, O) A CCM from a patient stained with H&E (N) or Prussian blue (O). Note similar appearance to the zebrafish lesion shown in (J, K). Arrow indicates dilated vessel. Scale bar: 50 μm. (P) CCMs were significantly reduced in *ccm2* CRISPR adult fish on *klf2a^-/-^* background compared to that on *klf2a^+/+^* background. Total number of embryos in each group is indicated. p=0.0076. Two-tailed Fisher's exact test was used for comparison.

The online version of this article includes the following figure supplement(s) for figure 7:

**Figure supplement 1.** *Ccm2* CRISPR adult fish displayed body wall lesions.

**Figure supplement 2.** Inhibiting Rho kinase blocks caudal venous plexus (CVP) cavernoma formation in *ccm2* CRISPR zebrafish.

## *Ccm2* mosaicism causes multi-cavernous malformations

Initially we ascribed the absence of segmental CVP dilation in *ccm2* null fish (*Mably et al., 2006*; *Renz et al., 2015*) solely to the reduced blood flow caused by the dilated heart. This explanation is insufficient because rescue of the heart phenotype in global *krit1 (ccm1)* null fish was not reported to cause segmental CVP dilation or CCMs (*Rödel et al., 2019*). Normal intussusceptive angiogenesis requires an orderly patterning of high and low flow signaling (*Karthik et al., 2018*). *Ccm2* mosaicism causes a random upregulation of *klf2a*, a key effector of flow signaling, thereby disrupting this orderly patterning of flow signaling. In contrast, the global knockout uniformly upregulates klf2a so that the patterning of other flow-sensitive signals can guide the completion of the intussusceptive arborization.

The dilated multi-cavernous CVP lesions described here resemble multi-cavernous CCM (*McDonald et al., 2011*) and their formation required mosaicism. Recent studies found that multi-cavernous murine CCMs are mosaic for inactivation of *Ccm3* (*Detter et al., 2018*; *Malinverno et al., 2019*). The mouse studies emphasized that simply dilated vessels are not mosaic and contained only *Ccm3* null endothelial cells (*Detter et al., 2018*; *Malinverno et al., 2019*). Mosaicism in multi-cavernous murine CCM was ascribed to recruitment of wild-type cells to the clonal CCM (*Detter et al., 2018*; *Malinverno et al., 2019*). Importantly, the mouse studies did not address the mechanism by which multi-cavernous lesions form. Here, we have shown that mosaicism is a prerequisite for formation of multi-cavernous CVP lesions because it disorganizes the flow signaling required for orderly sprouting and intussusceptive angiogenesis that remodel the CVP.

## *Ccm2* CRISPR zebrafish are an authentic CCM model

As in humans (*Akers et al., 2009*; *McDonald et al., 2011*), the fish CCM lesions arise as a consequence of mosaic inactivation of CCM genes. Second, as in humans, chronic bleeding leads to iron deposition; this finding contrasts with the lack of iron deposition seen in acute mouse CCM models (*Zeineddine et al., 2019*). Third, similar to humans, histologically typical lesions are distributed throughout the CNS in contrast to the hindbrain-restricted lesions in acute mouse models (*Zeineddine et al., 2019*). Fourth, as is true in mouse models (*Cuttano et al., 2016*; *Zheng et al., 2014*), the development of CCM depends on *klf2a*, the orthologue of murine *Klf2* and paralogue of *Klf4*, indicating that they form by the same pathogenetic mechanism. That said, in contrast to the KLF2 dependence of CVP dilation, injection of a KLF4 morpholino (*Li et al., 2011*) did not rescue this lesion (our unpublished data). There are chronic sensitized mouse models which do exhibit hemosiderin deposits and lesions throughout the CNS (*McDonald et al., 2011*); however, these models require cumbersome breeding schemes and mice of more than 3 months of age. That said, a recent report that postnatal induction of brain endothelial cell-specific ablation of the *Ccm2* gene using the inducible *Slco1c1*-CreER^T2^ mouse results in iron deposits around CCM throughout the murine brain at 3 months of age has great promise (*Cardoso et al., 2020*). In contrast to existing mouse models, the present model uses CRISPR-Cas9 to generate highly penetrant typical lesions throughout the CNS, requires about 2 months, and can be induced in mutant strains without additional breeding. Thus, this model should be a useful tool in future studies to assess the effect of the many genetic manipulations possible in zebrafish (*Gore et al., 2018*) on the pathogenesis of CCM

and to provide a complement to pharmacological screens directed at the dilated heart phenotype of *ccm1* or *ccm2* mutant fish (**Otten et al., 2018**).

In sum, the present work reveals a new embryonic vascular malformation, a multi-cavernous dilation of the CVP that resembles multi-cavernous Stage 2 CCM. The CVP malformation requires blood flow and mosaic inactivation of *ccm2* and is caused by abortive intussusceptive angiogenesis as a consequence of imbalanced flow signaling. The high penetrance and resemblance of the embryonic CVP malformation to multi-cavernous CCM suggest that it will be a useful phenotype for pharmacological or morpholino-based analyses. That said, the CVP does lack CNS accessory cells, such as astrocytes (**Lopez-Ramirez et al., 2021**), that promote CCM development. Indeed, we recently reported that propranolol blocks the embryonic CVP malformation by β1 adrenergic receptor antagonism (**Li et al., 2021**), a result that comports with the beneficial effects of propranolol in murine CCM models (**Li et al., 2021**; **Oldenburg et al., 2021**) and in anecdotal reports in humans (**Lanfranconi et al., 2020**; **Reinhard et al., 2016**). We have found that blockade or Rho kinase also ameliorates the CVP lesion (**Figure 7—figure supplement 2**) as it does murine CCM (**McDonald et al., 2012**). In addition, we report a tractable zebrafish model of CNS CCM that mimics the mammalian disease in mosaicism, lesion histology and distribution, and dependence on KLF2 transcription factors. A particularly appealing feature of these two new zebrafish models is that disease pathogenesis can be studied on mutant backgrounds without the need for additional breeding. Manipulations that ameliorate the embryonic lesion can then readily be tested for effects on the formation of brain CCMs that occur in the 50% of *ccm2* CRISPR embryos that survive to adulthood.

## Materials and methods

### Key resources table

| Reagent type (species) or resource | Designation | Source or reference | Identifiers | Additional information |
|---|---|---|---|---|
| Gene (*Danio rerio*) | ccm2 | http://www.ensembl.org/ | ENSDARG00000013705 | |
| Gene (*Danio rerio*) | klf2a | http://www.ensembl.org/ | ENSDARG00000042667 | |
| Strain, strain background (*Danio rerio*) | ccm2[m201] | zfin.org | ZDB-ALT-980203–523 | |
| Strain, strain background (*Danio rerio*) | klf2a[ig4] | zfin.org | ZDB-ALT-161103–5 | |
| Strain, strain background (*Danio rerio*) | Tg(fli1:EGFP)[y1] | zfin.org | ZDB-ALT-011017–8 | |
| Strain, strain background (*Danio rerio*) | Tg(gata1:dsred)[sd2] | zfin.org | ZDB-ALT-051223–6 | |
| Strain, strain background (*Danio rerio*) | Tg(klf2a:H2b-EGFP) | zfin.org | ZDB-ALT-161017–10 | |
| Strain, strain background (*Danio rerio*) | Tg(fli1:negfp)[y7] | zfin.org | ZDB-ALT-060821–4 | |
| Strain, strain background (*Danio rerio*) | Tg(kdrl:mcherry)[is5] | zfin.org | ZDB-ALT-110127–25 | |
| Recombinant DNA reagent | pCS2-nls-zCas9-nls | addgene.org | 47929 | |
| Recombinant DNA reagent | pT7-gRNA | addgene.org | 46759 | |

*Continued on next page*

*Continued*

| Reagent type (species) or resource | Designation | Source or reference | Identifiers | Additional information |
|---|---|---|---|---|
| Commercial assay or kit | mMESSAGE mMACHINE SP6 Transcription Kit | Thermo Fisher Scientific Wlatham, MA | AM1340 | |
| Commercial assay or kit | MEGAshortscript T7 Transcription kit | Thermo Fisher Scientific, Waltham, MA | AM1333 | |
| Sequence-based reagent | crRNA-1 | This paper | ccm2 gRNA | GGTGTTTCTGAAAGGGGAGA |
| Sequence-based reagent | crRNA-2 | This paper | ccm2 gRNA | GGAGAAGGGTAGGGATAAGA |
| Sequence-based reagent | crRNA-3 | This paper | ccm2 gRNA | GGGTAGGGATAAGAAGGCTC |
| Sequence-based reagent | crRNA-4 | This paper | ccm2 gRNA | GGACAGCTGACCTCAGTTCC |
| Chemical compound, drug | ccm2-MO | zfin.org | ZDB-MRPHLNO-060821–3 | GAAGCTGAGTAATACCTTAACTTCC |
| Chemical compound, drug | tnnt-MO | zfin.org | ZDB-MRPHLNO-060317–4 | CATGTTTGCTCTGATCTGACACGCA |
| Chemical compound, drug | gata1-MO | zfin.org | ZDB-MRPHLNO-050208–10 | CTGCAAGTGTAGTATTGAAGATGTC |
| Chemical compound, drug | tif1γ -MO | afin.org | ZDB-MRPHLNO-110321–1 | GCTCTCCGTACAATCTTGGCCTTTG |
| Chemical compound, drug | klf2a-MO | afin.org | ZDB-MRPHLNO-100610–8 | GGACCTGTCCAGTTCATCCTTCCAC |
| Chemical compound, drug | klf2b-MO | zfin.org | ZDB-MRPHLNO-150427–1 | AAAGGCAAGGTAAAGCCATGTCCAC |
| Software | Volocity | PerkinElmer Waltham, MA | Volocity | |
| Software | ZEN | Zeiss, Oberkochen, German | ZEN 2.3 SP1 | |
| Software | ImageJ software | ImageJ (http://imagej.nih.gov/ij/) | RRID:SCR_003070 | |
| Software | GraphPad Prism software | GraphPad Prism (https://graphpad.com) | Prism five for Windows | Version 5.01 |

## Zebrafish lines and husbandry

Zebrafish were maintained and with approval of Institutional Animal Care and Use Committee of the University of California, San Diego. The following mutant and transgenic lines were maintained under standard conditions: *ccm2^m201* (**Mably et al., 2006**), *klf2a^ig4* (**Steed et al., 2016**), *Tg(fli1:EGFP)^y1* (**Lawson and Weinstein, 2002**), *Tg(gata1:dsred)^sd2* (**Traver et al., 2003**), *Tg(fli1:negfp)^y7* (**Roman et al., 2002**), *Tg(klf2a:H2b-EGFP)* (**Heckel et al., 2015**), *Tg(kdrl:mcherry)^is5* (**Jin et al., 2005**), and *casper* (**White et al., 2008**). See Expanded Materials and Methods for morpholino injections. Morpholinos sequences are shown in *Supplementary file 1* (Morpholino sequences).

Plasmids pCS2-nls-zCas9-nls (47929) and pT7-gRNA (46759) were bought from Addgene. Crispr RNA (crRNA) sequences were listed in *Supplementary file 2* (crRNA sequences for zebrafish *ccm2*). Target gRNA constructs were generated as described before (**Jao et al., 2013**). PCS2-morangeccm2, pCS2-morangeccm2 mutant(L197R), pCS2-morangeklf2a, pCS2-morangeΔklf2a were cloned by infusion (Clontech) as follows: mOrange was cloned into ClaI, and linker sequence (5'-ggcagcgcgggcagcgcggcgggcagcggcgaattt-3') between ClaI and EcoRI. Then ccm2, L197R mutant, klf2a or Δklf2a sequence were cloned into EcoRI, respectively. These plasmids were then double-digested by SalI and NotI (NEB), and the fragment containing CMV promoter and coding sequence were purified and 0.5 nl of a 200 ng/μl solution was injected into single cell embryos. Primer sequences are listed in *Supplementary file 3* (Primers for template DNA synthesis).

## RNA synthesis

For cas9 mRNA, pCS2-nls-zCas9-nls was digested by NotI and then purified by column (Macherey-Nagel) as template. Capped nls-zCas9-nls RNA was synthesized using mMESSAGE mMACHINE SP6 Transcription Kit (Thermo Fisher Scientific) and purified through lithium chloride precipitation described in the same kit. For gRNA synthesis, gRNA constructs were linearized by BamHI digestion and purified by column (Macherey-Nagel). gRNA was synthesized by in vitro transcription using MEGAshortscript T7 Transcription kit (Thermo Fisher Scientific) and purified by alcohol precipitation described in the same kit. The concentration of nls-zCas9-nls RNA and gRNA were measured by NanoDrop 1000 Spectrophotometer (Thermo Fisher Scientific), and their quality was confirmed by electrophoresis through a 1% (wt/vol) agarose gel. The final concentrations for RNA injection are as follows: cas9 750 ng/µl, gRNA 120 ng/µl, and injection volume is 0.5 nl.

## Whole mount in situ hybridization

Zebrafish embryos were collected at 48 hpf and fixed with 4% paraformaldehyde overnight. In situ hybridization was performed as described before (*Thisse and Thisse, 2008*). The hybridization temperature is 68°C, and the probe concentration is 1 ng/µl. For primers used to amplify the template DNA for probe synthesis, see Expanded Materials and Methods. The images for in situ hybridization were captured by Olympus MVX10, Macro-view.

## Airyscan imaging and 3D reconstruction

Embryos for imaging were anesthetized with egg water containing 0.016% tricaine (3-amino benzoic acid ethyl ester, Sigma-Aldrich) and then embedded in 1% low melting point agarose (Invitrogen 16520050). Imaging was performed with Zeiss 880 Airyscan confocal under the standard Airyscan mode, and a 20×/NA 0.8 objective was used. Maximum projection was performed with ZEN (Zeiss). 3D reconstruction was performed with Volocity (PerkinElmer).

## Laser ablation of intravascular pillars

Laser ablation of intravascular pillars was performed using targeted ultrafast laser pulses that were generated with a multi-pass Ti:Al$_2$O$_3$ amplifier of local construction that followed a previously published design (*Nishimura et al., 2006*) and operated at a 5 kHz pulse rate. The ablation beam and the imaging beam were combined with a polarizing beamsplitter (*Nishimura et al., 2006*) prior to the microscope objective. The two beams were focused in the same focal plane and the ablation beam was centered in the area that is raster-scanned by the imaging beam so that ablation occurred at the center of the TPLSM imaging field. The energy per pulse of the ablation beam was tuned with neutral density filters and the quantity of pulses was controlled by a mechanical shutter (Uniblitz LS3Z2 shutter and VMM-D1 driver; Vincent). The energy and number of pulses was adjusted based on damage evaluated from the real-time TPLSM images and ranged between 0.2 and 0.4 µJ.

## Live imaging of endothelial pillar ablation

Live images of the fish vessels were obtained with a two-photon laser scanning microscope of local design (*Nishimura et al., 2006*), which was adapted to include an ablation beam. Low-energy, 100 fs, 76 MHz pulses for TPLSM were generated by a Titanium:Sapphire laser oscillator (Mira F-900; Coherent Inc) that was pumped by a continuous wave laser (Verdi V-10 Nd:YVO4 laser; Coherent Inc). The imaging laser pulses were scanned in a raster pattern by galvanometric mirrors that are relay-imaged to the rear aperture of the objective. The two-photon excited fluorescence is reflected by a dichroic mirror and transmitted to a photomultiplier tube. To produce laser pulses for ablation while imaging, we employed a Pockels cell (QS-3 with NVP-525D driver and DD1 timing circuit; Quantum Technologies) to reroute 1 in 76,000 pulses from the oscillator pulse train to seed a multi-pass Titanium:Sapphire amplifier that is pumped by a Q-switched laser (Corona; Coherent). A half-wave plate (λ/2) rotates the polarization of the amplified pulses to lie perpendicular to that of the laser oscillator and thus permits both the ablation beam and the imaging beam to be routed to the microscope objective with a polarizing beamsplitter. We used a 25×/NA 0.95, water immersion objective (Olympus) for imaging and ablation.

## Histology

Hematoxylin and eosin stain and Perl's Prussian blue stain were performed as described (*Zeineddine et al., 2019*).

## Zebrafish brain dissection, CUBIC treatment, and light sheet imaging

Zebrafish brain dissection was performed as previously described (*Gupta and Mullins, 2010*). CUBIC was optimized on the basis of previous report (*Susaki et al., 2015*). The brains were fixed with pH 7.5 4% PFA for 24 hr and then washed with PBS for 24 hr. After PBS wash, CUBICR1 treatment was then performed at 37°C in water bath for 42 hr. Samples were imaged in CUBICR2 as medium with ZEISS Lightsheet Z.1. Scanning was performed with 5× dual illumination optics and 5× objective.

## Statistical analysis

Statistical analysis was performed with GraphPad Prism. p-Values were calculated by paired two-tailed Student's t-test unless otherwise specifically indicated. The mean and SD were shown in the bar graphs.

## Acknowledgements

We gratefully acknowledge Brant Weinstein for sharing a CUBIC protocol, David Traver, Miguel Lopez-Ramirez, Alexandre Gingras, and Sara McCurdy for valuable discussion and criticism, and Jennifer Santini and Marcy Erb for microscopy technical assistance. We also acknowledge resources provided by the UCSD School of Medicine Microscopy Core (NINDS P30 NS047101).

## Additional information

### Funding

| Funder | Grant reference number | Author |
|---|---|---|
| National Heart, Lung, and Blood Institute | HL 139947 | Mark H Ginsberg |
| National Institutes of Health | NS 92521 | Thomas Moore<br>Rhonda Lightle<br>Issam A Awad<br>Mark H Ginsberg |
| National Institute of Mental Health | R35 NS097265 | David Kleinfeld |
| National Institutes of Health | R01 NS108472 | Iftach Shaked |
| Be Brave for Life | | Wenqing Li |

The funders had no role in study design, data collection and interpretation, or the decision to submit the work for publication.

### Author contributions

Wenqing Li, Conceptualization, Data curation, Formal analysis, Investigation, Methodology; Virginia Tran, Belinda Xue, Thomas Moore, Rhonda Lightle, Investigation; Iftach Shaked, David Kleinfeld, Investigation, Methodology; Issam A Awad, Formal analysis; Mark H Ginsberg, Conceptualization, Data curation, Formal analysis, Funding acquisition, Project administration

### Author ORCIDs

David Kleinfeld (iD) https://orcid.org/0000-0001-9797-4722
Mark H Ginsberg (iD) https://orcid.org/0000-0002-5685-5417

### Ethics

Animal experimentation: This study was performed in strict accordance with the recommendations in the Guide for the Care and Use of Laboratory Animals of the National Institutes of Health. All of the animals were handled according to approved institutional animal care and use committee (IACUC) protocols (#S14135 ) of the University of California San Diego.

### Decision letter and Author response

Decision letter https://doi.org/10.7554/eLife.62155.sa1
Author response https://doi.org/10.7554/eLife.62155.sa2

## Additional files

### Supplementary files

- Supplementary file 1. Morpholino sequences.
- Supplementary file 2. crRNA sequence for zebrafish ccm2.
- Supplementary file 3. Primers for template DNA synthesis.
- Transparent reporting form

### Data availability

Raw phenotype counts have been provided in figures and figure legends.

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
