## [Decision Letter]

Thank you for submitting your article "Abortive Intussusceptive Angiogenesis Causes Multi-Cavernous Vascular Malformations" for consideration by *eLife*. Your article has been reviewed by 4 peer reviewers, one of whom is a member of our Board of Reviewing Editors, and the evaluation has been overseen by Edward Morrisey as the Senior Editor. The following individuals involved in review of your submission have agreed to reveal their identity: Victoria L Bautch (Reviewer #2); Brent Derry (Reviewer #3).

The reviewers have discussed the reviews with one another and the Reviewing Editor has drafted this decision to help you prepare a revised submission.

All the reviewers agree that the paper has potential but they also ask for relevant revision. More specifically:

1. A detailed discussion on the working hypothesis on the role of flow and mosaicism on CCM lesion development.

2. The use of the fish model presented here for a high throughput screening of thousands of drugs looks indeed a very difficult if not impossible task. Would the authors be able to answer to this criticism?

3. Statistic is also poor and, in many cases, missing. This is a crucial aspect of the study and should be better reported and described.

The paper therefore warrants publication in *eLife* but revision is needed. Please see the full reviews below for further comments.

*Reviewer #1:*

Li et al. present a new model of CCM in Z.fish using mosaic inactivation of ccm2 through Crisp technique. Vascular malformations develop in the Caudal venous plexus in the embryos and in the central nervous system of the embryos surviving to adulthood.

These malformations in the Caudal venous plexus form for aberrant intussusception and depend on flow, accumulation of erythrocytes and mosaic upregulation of klf2.

The authors propose this model for large pharmacological screening of CCM phenotype-correcting drugs. The first step would test compounds on malformations in the Caudal venous plexus of Z. fish embryos. The following validation step would test the malformations in the CNS of the mutant adults.

The model is skillfully presented. However, the interpretation of the mechanism is not fully supported by the experimental data which appear often more suggestive than conclusive.

1. The morphological features and some of the mechanisms directing the formation of vascular malformations in Z. fish embryos are studied in details, while those in the central nervous system of adults are only shown to depend on the expression of klf2. To which extent is the mechanism driving the lesions in the Caudal venous plexus modeling that in the CNS?

Are the lesions in the CNS depending on blood flow and erythrocyte accumulation and do they show abortive intussusception as in the Caudal venous plexus? In addition, do vascular malformations in the Caudal venous plexus show increased permeability and hemorrhages as those in the CNS? Organ-specific microenvironment strongly influences endothelial responses. Therefore, the issues above should be defined for comprehensively describe the biology of the model and for supporting the validity of the two-step screening proposed.

Most importantly, the limits of the intussusceptive mechanism of lesion formation in Z. fish Caudal venous plexus as a model for human cavernomas in the CNS are neither tested nor demonstrated.

2. While the advantages of using Z. fish for direct and rapid in vivo analysis of CCM lesions is appealing some caveats are evident. Is Z. fish equally sensitive to mosaic deletion of ccm1 and ccm3 as to ccm2? The literature about the effects of mutation of CCM genes in Z. fish, well summarized by the authors, indicates that Z. fish could react in a peculiar way to the mutation of different CCM genes. This can limit the use of Z. fish as a model of human cavernomas.

3. 'Mosaic upregulation of KLF2a is sufficient for cavernoma formation in CVP'.

Mosaic upregulation of klf2 induces malformation in the Caudal venous plexus in 6% of the embryos. This is a small percentage compared to 30% ccm2 Crisp embryos developing malformations in Caudal venous plexus. This different efficiency should be explained.

Is the level of overexpressed klf2 in the range reached by klf2 after ccm2 deletion? Where is this overexpressed klf2 actually localized? Is this overexpressed klf2 localized in the pillars of the vascular malformation? Is the endogenous klf2 upregulated in the pillars of the vascular malformations of ccm2 Crisp embryos? This is not visible in Figure 4D. In addition, is the mosaic increase of klf2 able to induce malformations in the CNS in the adult?

4. Do the malformations contain ccm2 null endothelial cells? No direct evidence is presented, besides the rescuing of the dilated CVP phenotype by ccm2-mRNA. Do the lesion-free areas of the Caudal venous plexus contain equal density of ccm2 null endothelial cells as in lesion areas?

5. Statistical analysis needs be shown to support the reproducibility of the data presented in Figure 3 E and F. Which was the range of reduction of vein diameter after pillar ablation and how many embryos were used to reproduce this result? This aspect needs to be strengthened has much of the model interpretation is based on the role of intraluminal pillar in obstructing blood flow and causing vessel dilation.

6. In several figures presenting morphologic data statistical analysis is missing and should be added. Figure 6A, B, C quantification of the immunofluorescence results is lacking. How many endothelial cells show upregulation of klf2 in ccm2 Crisp? No control of Figure 6B is shown.

7. How would erythrocytes contribute to the formation/dilation of the cavernae? Would this be a mechanical effect or would erythrocytes convey other signals to endothelial cells? Are erythrocytes present in the cavernae ccm2 null?

8. It is not clear what come first: both erythrocyte null and heart-silenced ccm2 crisp show reduction of dilated CVP. Do erythrocytes circulate in silenced heart Z. fish? In addition, how is klf2 regulated in erythrocytes null and heart-silenced ccm2 Crisp Z. fish?

9. Are the embryos developing malformations in the CVP surviving? If yes, are the cavernoma in the Caudal venous plexus persisting?

10. When and how do the cavernomas form in the brain of the surviving fishes? This is a significant aspect to define in this model, as cavernomas in the central nervous system are the malformations with pathological consequences in humans. How long do these mutant adults survive?

11. In murine models klf4 is also required for cavernoma formation. Is the same true for Z. fish?

*Reviewer #2:*

The paper by Li et al. investigates the effects of mosaic manipulation of CCM2 in zebrafish embryos and adult fish, and describes a CVP dilation linked to intussusceptive angiogenesis in embryos and neurovascular lesions in adult fish. The primary finding is that mosaic deletion of CCM2 leads to differences in flow-mediated responses of EC that lead to the embryonic phenotype, and that it occurs in the context of intussusceptive angiogenesis. These findings are well-supported by genetic, morpholino (MO) and pharmacological analysis and overall careful and rigorous analysis. The novelty is substantial in both findings and experimental approach (CRISPR/Cas induced mosaicism) and provides explanations for some of the disease phenotypes. However, there are some issues that, if addressed, would substantially improve the work:

1. I understand why the adult phenotype is presented, and it does show validity of the adult fish as a model. However, in terms of mechanism, it raises interesting questions that were not adequately addressed – for example, how do lesions form in a tissue that is not known to undergo intussusceptive angiogenesis? Is Klf2 expression also mosaic in the adult fish brains? Can the fish be used to generate mosaic Klf2 over-expression and determine effects in the adult independent of CCM2 manipulation?

2. Many of the statements regarding the data in the Results and Discussion are stated as facts rather than presented as conclusions – there are too many to enumerate, but examples: p. 18: "first zebrafish model of CCM"; p. 19: "pillars failed…to split due to mosaic over-expression of klf2a….". The work is very rigorous but all experiments have caveats. The Discussion is also very focused on why the adult fish is a good model for clinical CCM, and many interesting aspects of the bulk of the work presented in the embryo are not addressed. For example, why is mosaic loss but not global loss of flow-sensing proposed to lead to the phenotype? Are the mechanisms the same in vessel beds that do not undergo intussusceptive angiogenesis? What is the effect of the CCM complex vs. CCM2 alone?

3. The work is quite novel and exciting; however, it is difficult to keep track of the different manipulations and combination of manipulations, and this is exacerbated by very poor labeling of figures and descriptions in figure legends. Many of the Y-axis labels merely say "% phenotype" with no context for complex combinations of manipulations. Images are not well labeled for stains/reporters. There is no documentation that most experiments were mosaic for deletion, which is central to the model put forward – the labels suggest global LOF. Suggest use (or figure out if this is new) a nomenclature for mosaic deletion and use consistently. Please be clear about GOF vs. LOF manipulations.

*Reviewer #3:*

This manuscript describes a mosaic model of ccm2 deletion in zebrafish. The authors report defects in the vasculature including dilations in the caudal venous plexus (CVP) and cranial vessels (CV), as well as previously described vascular and heart defects. Confocal imaging and 3D reconstruction showed defective lumenization of endothelial pillars, resulting in multiple chambers that accumulate blood and suggest incomplete intussusceptive angiogenesis. Laser ablation of defective pillars relieved dilation and restored blood flow, suggesting that the pillars caused dilated CVP and flow defects. They demonstrate that blood flow is required for the dilation of CVP and intussusceptive pillar formation in mosaic ccm2 mosaics, which casts doubt on a recent study showing that blood flow actually suppresses vascular anomalies in zebrafish harboring a germline krit1 (CCM1) knockout (PMID: 31495257). Furthermore, they show that erythrocyte accumulation in defective CVP drives their dilation. They convincingly demonstrate that mosaicism accounts for CVP dilation by co-injection of a sublethal dose of ccm2 morpholino with the CRISPR mix. They go on to show that upregulation of the Klf2 transcription factor, which acts downstream of Ccm2, accounts for dilation of CVP in ccm2 mosaic mutants. This impedes the flow signaling required for intussusceptive angiogenesis that remodels the CVP and likely explains how these lesions form in human CCM patients. Finally, ccm2 mosaic fish that did not exhibit vascular anomalies eventually develop lesions in their brain vasculature and spinal cords by adulthood. This study shows that mosaicism is a pre-requisite for formation of multi-cavernous lesions and provides the first zebrafish model that accurately recapitulates the disease in humans. This is an important advance in our understanding of the genesis of CCM lesions that should be suitable for publication after a few concerns are addressed.

1. Do actin stress fibers form and/or does pMLC increase in endothelial cells of lesions? This would highlight conservation of CCM lesion mechanisms between fish and human.

2. Since the authors have previously shown that inhibition of Rho kinase can suppress lesion formation in mouse models it would be nice to see if this is also true in their mosaic zebrafish model.

3. While the authors show that mosaic overexpression of klf2a is responsible for the formation of vascular defects in the CVP of ccm2 mosaic fish, they do not show this when Ccm2 is overexpressed (Figure 5B). Therefore, they should inject linearized ccm2 fused to mOrange into the klf2a mutants to see if these embryos also fail to develop CVP dilations.

4. In the text the authors state that CCM lesions were not observed when ccm2 was edited in klf2a-/- mutants, but in Figure 7P they report 1/10 embryos with lesions. The text should be amended to reflect this result, as it misrepresents their conclusions.

5. There has been a bit of controversy in the zebrafish community regarding the use of "Crispants" and morpholinos versus germline mutants that the authors should acknowledge (ie PMID: 32968253). I have no issues with the interpretation of data in this study since they performed rescue experiments but given the differences in phenotypes compared with germline mutants this needs to be discussed.

*Reviewer #4:*

Wenqing Li et al. introduce a novel mechanism that may cooperate in the formation of malberry vascular development in CCM2 deficient Zebra fish. These authors claim that in the caudal venous plexus, mosaic inactivation of CCM2 together with a patchy upregulation of klf2a results in the formation of pillars that create a partial obstruction of the blood flow due to red cell accumulation in the lumen. Morphologically, the pillars mimic intussusceptive angiogenesis and this alters the correct development of the vasculature. In CCM deficient fish the pillars are unable to fully cross the lumen and create multi-cavernous malberry-like malformations. Overall, these morphological observations are of interest and introduce partially novel concepts.

However:

– I am not convinced that this model is better than the mouse models available. It is a complex, time limited and variable condition. The percentage of fish resulting affected is relatively low and this prevents the use of this model for high throughput screening of thousands of compounds, as proposed by the authors.

– Not all the conclusions are substantiated by previous work in the mouse. For instance, the authors underline that klf2a is the major effector of cavernoma formation in the fish, while in mice klf4 is equally or even more important.

– Most importantly, there are no data showing that the formation of pillars and abortive angiogenesis also occur in CCM2 deficient mammals.

---

## [Author Response]

Reviewer #1:[…] The model is skillfully presented. However, the interpretation of the mechanism is not fully supported by the experimental data which appear often more suggestive than conclusive.1. The morphological features and some of the mechanisms directing the formation of vascular malformations in Z. fish embryos are studied in details, while those in the central nervous system of adults are only shown to depend on the expression of klf2. To which extent is the mechanism driving the lesions in the Caudal venous plexus modeling that in the CNS?

Our data show that both CNS and CVP lesions arise following mosaic inactivation of a CCM gene and depend on *klf2a.* We agree that there are important differences between the environment of the brain and the CVP and have inserted the following comment to emphasize this point: “That said, the CVP does lack CNS accessory cells, such as astrocytes,(Lopez-Ramirez et al., 2021) that promote CCM development.” We have also cited a recently published report that development of the CVP lesion, like the brain CCM, is inhibited by propranolol. Our previous work showed that blocking Rho Kinase would decrease CCM in a mouse model and, in an experiment suggested by referee 3, we now show that blocking Rho Kinase inhibits the CVP lesion (Figure 7—figure supplement 2).

Are the lesions in the CNS depending on blood flow and erythrocyte accumulation and do they show abortive intussusception as in the Caudal venous plexus? In addition, do vascular malformations in the Caudal venous plexus show increased permeability and hemorrhages as those in the CNS? Organ-specific microenvironment strongly influences endothelial responses. Therefore, the issues above should be defined for comprehensively describe the biology of the model and for supporting the validity of the two-step screening proposed.Most importantly, the limits of the intussusceptive mechanism of lesion formation in Z. fish Caudal venous plexus as a model for human cavernomas in the CNS are neither tested nor demonstrated.

Addressing this concern would require imaging the human disease at high resolution as it develops, which is presently not technically feasible.

2. While the advantages of using Z. fish for direct and rapid in vivo analysis of CCM lesions is appealing some caveats are evident. Is Z. fish equally sensitive to mosaic deletion of ccm1 and ccm3 as to ccm2? The literature about the effects of mutation of CCM genes in Z. fish, well summarized by the authors, indicates that Z. fish could react in a peculiar way to the mutation of different CCM genes. This can limit the use of Z. fish as a model of human cavernomas.

We have not been able to identify candidate guide RNAs for *ccm3.* Five candidate *ccm1* guide RNAs (Reviewer Table) failed to produce sufficient indels. We were therefore unable to do these experiments; however, we note that CCM1 and CCM2 function as a complex and the phenotypic effects of their loss in mammals and zebrafish have been indistinguishable.

3. 'Mosaic upregulation of KLF2a is sufficient for cavernoma formation in CVP'.Mosaic upregulation of klf2 induces malformation in the Caudal venous plexus in 6% of the embryos. This is a small percentage compared to 30% ccm2 Crisp embryos developing malformations in Caudal venous plexus. This different efficiency should be explained.

The degree and sites of mosaicism in the KLF2a over expression and *ccm2* CRISPR experiments are random. Similarly, the abundance of over-expressed KLF2a per cell is also random. Thus, frequencies of CVP dilation can vary between the two approaches.

Is the level of overexpressed klf2 in the range reached by klf2 after ccm2 deletion?

As mentioned above, there is considerable variability in the quantity of KLF2a over-expressed in each cell. Furthermore, there is no easy way to compare the over-expression of mOrange-KLF2a with the increase in KLF2a promoter-driven GFP expression in the *ccm2* CRISPR experiment.

Where is this overexpressed klf2 actually localized? Is this overexpressed klf2 localized in the pillars of the vascular malformation? Is the endogenous klf2 upregulated in the pillars of the vascular malformations of ccm2 Crisp embryos?

We have no access to an antibody against fish KLF2a that could be used for this purpose.

4. Do the malformations contain ccm2 null endothelial cells? No direct evidence is presented, besides the rescuing of the dilated CVP phenotype by ccm2-mRNA. Do the lesion-free areas of the Caudal venous plexus contain equal density of ccm2 null endothelial cells as in lesion areas?

In the transient over-expression of KLF2a and in *ccm2* CRISPR experiments, we did not have antibodies available to visualize the CCM2 and KLF2.

5. Statistical analysis needs be shown to support the reproducibility of the data presented in Figure 3 E and F. Which was the range of reduction of vein diameter after pillar ablation and how many embryos were used to reproduce this result? This aspect needs to be strengthened has much of the model interpretation is based on the role of intraluminal pillar in obstructing blood flow and causing vessel dilation.

We report: “In 3 such independent experiments, severing these pillars resulted in a 29± 4% reduction in vessel diameter (p=0.0004, two-tailed T test).”

6. In several figures presenting morphologic data statistical analysis is missing and should be added. Figure 6A, B, C quantification of the immunofluorescence results is lacking. How many endothelial cells show upregulation of klf2 in ccm2 Crisp? No control of Figure 6B is shown.

We have added statistical analysis throughout the paper. In response to referee 2 we have relocated the KLF2a reporter data to Figure 4 in the revised paper. In Figure 4F we now show a quantitative analysis of reporter expression that documents the mosaic upregulation of KLF2a in *ccm2* CRISPR fish relative to controls.

7. How would erythrocytes contribute to the formation/dilation of the cavernae? Would this be a mechanical effect or would erythrocytes convey other signals to endothelial cells? Are erythrocytes present in the cavernae ccm2 null?

Nucleated erythrocytes, ~8μm in diameter, are trapped in the meshwork of intussusceptive pillars, wherein plasma can still circulate. In addition, by increasing the viscosity of blood, erythrocytes can contribute to the shear forces that drive intussusception.

8. It is not clear what come first: both erythrocyte null and heart-silenced ccm2 crisp show reduction of dilated CVP. Do erythrocytes circulate in silenced heart Z. fish?

When the heart is stopped, blood circulation ceases.

9. Are the embryos developing malformations in the CVP surviving? If yes, are the cavernoma in the Caudal venous plexus persisting?

We note (e.g. Abstract line 3) that we are describing a “novel lethal multi-cavernous lesion in the embryonic caudal venous plexus (CVP).”

10. When and how do the cavernomas form in the brain of the surviving fishes? This is a significant aspect to define in this model, as cavernomas in the central nervous system are the malformations with pathological consequences in humans. How long do these mutant adults survive?

Lesions form in the brain by 6 weeks post fertilization. Unfortunately, at the stage, the zebrafish are no longer transparent so we cannot easily observe the process in real time.

11. In murine models klf4 is also required for cavernoma formation. Is the same true for Z. fish?

We found that a published KLF4 morpholino did not prevent CVP dilation in *ccm2* CRISPR fish (Reviewer Figure) and mention this result in the discussion.

Reviewer #2:[…] 1. I understand why the adult phenotype is presented, and it does show validity of the adult fish as a model. However, in terms of mechanism, it raises interesting questions that were not adequately addressed – for example, how do lesions form in a tissue that is not known to undergo intussusceptive angiogenesis? Is Klf2 expression also mosaic in the adult fish brains? Can the fish be used to generate mosaic Klf2 over-expression and determine effects in the adult independent of CCM2 manipulation?

We agree that the question of whether the mechanism we have observed in the CVP occurs in the brain is of great interest. That said, because we cannot visualize development of the brain lesions in real time, we cannot establish this point. Similarly, generating a mosaic KLF2a expressing adult fish would be a useful experiment. That said, since the loss of *klf2a*, completely inhibited adult CCM formation, this time consuming experiment is not urgent and is beyond the present scope.

2. Many of the statements regarding the data in the Results and Discussion are stated as facts rather than presented as conclusions – there are too many to enumerate, but examples: p. 18: "first zebrafish model of CCM".

We have removed any reference to “the first” (e.g. in the Abstract).

p. 19: "pillars failed…to split due to mosaic over-expression of klf2a….". The work is very rigorous but all experiments have caveats. The Discussion is also very focused on why the adult fish is a good model for clinical CCM, and many interesting aspects of the bulk of the work presented in the embryo are not addressed. For example, why is mosaic loss but not global loss of flow-sensing proposed to lead to the phenotype?

We have addressed this important question at several point in the Discussion with respect to perturbed flow signaling and with respect to mosaicism.

Are the mechanisms the same in vessel beds that do not undergo intussusceptive angiogenesis? What is the effect of the CCM complex vs. CCM2 alone?

We indirectly addressed this issue by showing that over-expression of wild type *ccm2* but not *ccm2(L197R)* cause CVP dilation (Figure 6B and B’). CCM2(L197R) does not bind KRIT1 and is therefore not incorporated into the CCM complex.

3. The work is quite novel and exciting; however, it is difficult to keep track of the different manipulations and combination of manipulations, and this is exacerbated by very poor labeling of figures and descriptions in figure legends. Many of the Y-axis labels merely say "% phenotype" with no context for complex combinations of manipulations. Images are not well labeled for stains/reporters. There is no documentation that most experiments were mosaic for deletion, which is central to the model put forward – the labels suggest global LOF. Suggest use (or figure out if this is new) a nomenclature for mosaic deletion and use consistently. Please be clear about GOF vs. LOF manipulations.

Thank you for this comment. We have revised the paper and changed all of the ordinates to clearly state the phenotype..

Reviewer #3:[…]1. Do actin stress fibers form and/or does pMLC increase in endothelial cells of lesions? This would highlight conservation of CCM lesion mechanisms between fish and human.2. Since the authors have previously shown that inhibition of Rho kinase can suppress lesion formation in mouse models it would be nice to see if this is also true in their mosaic zebrafish model.

We now report (Figure 7—figure supplement 2) that, like mammalian CCM, inhibiting Rho kinase blocks development of the zebrafish CVP lesion.

3. While the authors show that mosaic overexpression of klf2a is responsible for the formation of vascular defects in the CVP of ccm2 mosaic fish, they do not show this when Ccm2 is overexpressed (Figure 5B). Therefore, they should inject linearized ccm2 fused to mOrange into the klf2a mutants to see if these embryos also fail to develop CVP dilations.

We report that there is a marked reduction CVP dilation when ccm2 is over expressed in klf2a mutants (Figure 6—figure supplement 1)

4. In the text the authors state that CCM lesions were not observed when ccm2 was edited in klf2a-/- mutants, but in Figure 7P they report 1/10 embryos with lesions. The text should be amended to reflect this result, as it misrepresents their conclusions.

Done.

5. There has been a bit of controversy in the zebrafish community regarding the use of "Crispants" and morpholinos versus germline mutants that the authors should acknowledge (ie PMID: 32968253). I have no issues with the interpretation of data in this study since they performed rescue experiments but given the differences in phenotypes compared with germline mutants this needs to be discussed.

We agree that this is an important issue and stress that the *klf2a* loss of function experiments were performed on both mutants and morphants. Secondly, the CVP dilation phenotype was rescued by re-expression of CCM2 (Figure 1G), thus controlling for off target effects of CRISPR.

Reviewer #4:[…] – I am not convinced that this model is better than the mouse models available. It is a complex, time limited and variable condition. The percentage of fish resulting affected is relatively low and this prevents the use of this model for high throughput screening of thousands of compounds, as proposed by the authors.

We have now modified the final paragraph of the discussion to remove any implication that this phenotype could be used for high throughput screening while emphasizing its potential utility in targeted genetic or pharmacological analyses. We suggest that the fish model has unique virtues as argued in the last paragraph of the discussion.

– Not all the conclusions are substantiated by previous work in the mouse. For instance, the authors underline that klf2a is the major effector of cavernoma formation in the fish, while in mice klf4 is equally or even more important.

We have not been able to substantiate a role for KLF4 in the zebrafish CVP lesion (Reviewer Figure) and have so stated in the paper. We were unaware of data showing that KLF4 in more important than KLF2 in the mammalian CCM or that either was essential for human CCM.

– Most importantly, there are no data showing that the formation of pillars and abortive angiogenesis also occur in CCM2 deficient mammals.

True, since the paper is describing zebrafish models.